# Enhanced energy storage in high-entropy superparaelectrics via local ferroelectric polarization

Tongxin Wei [1], Jinzhu Zou [1] ✉, Miao Song [1], Kai Zhu[1], Zhongna Yan[2], Zhifang Zhou [1], Xuefan Zhou[1], Kechao Zhou[1], Shujun Zhang [3] ✉ & Dou Zhang [1] ✉

Dielectric ceramic capacitors with ultrahigh power density have become essential in modern power electronics. Guided by phase-field simulations and experiments, we propose a "local ferroelectric–global superparaelectric" strategy. This approach enhances $P_m$ by introducing local ferroelectric polarization within a superparaelectric matrix, enabling superior energy storage performance. Introducing strong ferroelectric $PbTiO_3$ into a $(Bi_{0.2}Na_{0.2}K_{0.2}La_{0.2}Sr_{0.2})Ti_{0.9}Zr_{0.1}O_3$ high-entropy superparaelectric achieves an ultrahigh energy storage density of ~21 J/cm$^3$ with an efficiency of ~87% at 110 kV/mm. Multiscale structural characterization and theoretical calculations reveal the atomic-scale mechanism for this performance enhancement. At ≤ 30% $PbTiO_3$, the $Pb^{2+}$ lone pair effect is locally confined, boosting local ferroelectric distortion while maintaining a superparaelectric average structure for superior energy storage. At 40-50%, this effect extends throughout the matrix, inducing submicro-scale domains and macroscopic piezoelectricity. This work presents a design and material system for high-performance energy storage ceramics, laying the theoretical foundation for advanced high-entropy ferroelectric applications.

Dielectric ceramics are widely used as key components in pulsed power systems, such as pulse-forming networks, high-voltage pulse generators, and electromagnetic launch systems, due to their rapid charge–discharge rates and excellent stability[1–3]. However, conventional dielectric ceramics generally exhibit relatively low energy storage density, typically with recoverable energy storage density ($W_{rec}$) below 10 J/cm$^3$[4–7]. This limitation severely restricts their application in modern electronic devices, which are trending toward miniaturization, lightweight design, and integration. Therefore, effectively enhancing the energy storage performance of dielectric ceramics has become a significant challenge in current research on functional materials.

In recent years, the high-entropy strategy, as an emerging material design concept, has been introduced into dielectric ceramic systems,

demonstrating remarkable performance advantages[8–10]. In both ferroelectric thin films and ceramics, the introduction of high-entropy strategies has been demonstrated to significantly enhance energy storage density and efficiency[11–19]. The underlying mechanism primarily lies in the coexistence of multiple dissimilar atoms in the same lattice position within the high-entropy system. These cations possess distinct ionic radii, valences, and electronic configurations, which generate significant local compositional disorder, random fields, and lattice distortions, disrupting the long-range ferroelectric order and promote the formation of polar nanoregions (PNRs)[20–25]. This structural feature facilitates rapid and sufficient polarization response under an applied electric field, thereby enhancing energy storage capacity. It is noteworthy that the recoverable energy storage density

[1]State Key Laboratory of Powder Metallurgy, Central South University, Changsha, China. [2]School of Energy and Power Engineering, Changsha University of Science and Technology, Changsha, China. [3]Department of Chemistry, Department of Materials Science and Engineering, City University of Hong Kong, Kowloon, China. ✉e-mail: jinzhuzou@csu.edu.cn; s.j.zhang@cityu.edu.hk; dzhang@csu.edu.cn

($W_{rec}$) of dielectric ceramics is essentially a function of polarization intensity ($P$) varying with the external electric field ($E$), which can be calculated by the formula[26]:

$$W_{rec} = \int_{P_r}^{P_m} E dP \qquad (1)$$

where $P_m$ is the maximum polarization intensity, and $P_r$ is the residual polarization intensity. Therefore, to achieve a high $W_{rec}$, it is necessary to maintain a low $P_r$ while obtaining a high $P_m$, meaning the material should possess a large polarization difference ($P_m - P_r$). However, in high-entropy ceramic systems, multi-element doping often drives the system into a superparaelectric (SPE) state, which contains numerous non-polar regions. While this is beneficial for reducing $P_r$, it often limits the enhancement of $P_m$ (as shown in Figs. 1a1–a3), making it difficult to achieve sufficiently high polarization intensity under high electric fields. This has become a key scientific issue restricting further breakthroughs in the energy storage performance of high-entropy ceramics[1,12,27].

Guided by phase-field simulations and experimental evidence, this study systematically analyzed the influence of introducing para-electric, antiferroelectric, and ferroelectric phases into a superpara-electric state on polarization response. As illustrated in Fig. 1b, it reveals that incorporating ferroelectric phase into the superpara-electric high-entropy matrix can achieve a significant enhancement in maximum polarization ($P_m$) while effectively suppressing remanent polarization ($P_r$). The key to this performance improvement lies in the coupling interaction between the superparaelectric matrix and the locally strong ferroelectric (FE) phases. Specifically, as shown in Fig. 1c1, influenced by the surrounding superparaelectric environment, the locally ferroelectric phases do not exhibit macroscopic long-range order. Instead, while maintaining a high polarization magnitude, their polarization directions become disordered, resulting in a low macro-scopic net polarization. As depicted in Figs. 1c2–c5, under an external electric field, the flexible polarization response of the superpara-electric matrix significantly enhances the polarization switching within the strong polar regions, enabling macroscopic alignment of the strong polarizations and leading to increased saturation polarization. During field reduction, the polarization of the superparaelectric matrix also promotes the reversal of polarization in the strong polar regions, resulting in low remanent polarization. As shown in Fig. 1a4, based on this coupling mechanism, it is anticipated that the saturation polar-ization of the superparaelectric material can be enhanced, thereby improving its energy storage density. Building on this understanding, the research introduced the strongly ferroelectric component PbTiO₃ into Bi$_{0.2}$Na$_{0.2}$K$_{0.2}$La$_{0.2}$Sr$_{0.2}$Ti$_{0.9}$Zr$_{0.1}$O₃ (BNKLSTZ) high-entropy cera-mics, achieving energy storage performance: a recoverable energy storage density of approximately 21 J/cm³ and an energy storage effi-ciency ($\eta$) of about 87%, positioning it among the most advanced performance levels currently reported. Density functional theory analysis reveals that the presence of the lone pair electrons on Pb is responsible for generating the localized strong ferroelectric polariza-tion, providing the atomic-scale mechanism for the performance enhancement[28]. At lower PbTiO₃ content (≤30%), the localized lone pair effect of Pb²⁺ enhances local ferroelectric distortion while main-taining an average superparaelectric structure for excellent energy storage; as the content increases to 40–50%, this effect percolates throughout the matrix, leading to the emergence of submicron fer-roelectric domains and macroscopic piezoelectricity. This work not only provides a design strategy and an effective material system for high-performance energy storage ceramics but also lays a theoretical foundation for further research on high-entropy ferroelectrics in applications such as electrostrain and piezoelectricity.

## Results and discussion
### Phase field simulations
To investigate the influence of different components on the polar-ization response of the high-entropy superparaelectric matrix, we conducted phase-field simulations. For the high-entropy super-paraelectric matrix, based on the model proposed by Yang et al.[29], elements were treated as point defects with lowered Curie tem-peratures to simulate the refinement of domain structures due to multi-element incorporation (Supplementary Notes 1.1 and Figs. S1–S2). Because the high-entropy design results in a $T_m$ well below room temperature, the entire system exhibits a weakly polarized state in the absence of an external electric field (Fig. 2a1). When a positive electric field is applied, the field drives the reor-ientation and extension of polarization, leading to a pronounced polar state (Fig. 2a2). Conversely, under a negative electric field, polarization can also respond rapidly (Fig. 2a3). As discussed pre-viously, the disruption of polar order due to high-entropy effects results in a relatively low $P_m$ (Fig. 2a4). To examine the impact of different components within the superparaelectric matrix on polarization response, we performed further phase field simulations introducing paraelectric, antiferroelectric, and ferroelectric phases (Supplementary Note 1.2 and Fig. S3).

(i) For the paraelectric phase, which lacks spontaneous polariza-tion, the local polarization vanishes in the absence of an external electric field, as shown in Fig. 2b1. Under a positive/negative electric field, as illustrated in Figs. 2b2–b3, the polarization in the superpara-electric matrix responds, whereas the paraelectric region remains unresponsive. This leads to a significant reduction in saturated polar-ization in the $P–E$ hysteresis loop, thereby lowering the energy storage density of the system (Fig. 2b4).

(ii) For the antiferroelectric phase, antiparallel spontaneous polarizations exist internally, canceling each other out macro-scopically. As shown in Fig. 2c1, the whole system exhibits a weakly polarized state without an external electric field. Under an applied electric field, the antiparallel spontaneous polarizations in the anti-ferroelectric region become aligned, showing a higher polarization intensity than the superparaelectric matrix (Fig. 2c2). Consequently, the polarization magnitude in the local antiferroelectric region is lar-ger. When a reverse electric field is applied, the polarization in the antiferroelectric region can be reoriented, demonstrating a relatively high polarization strength (Fig. 2c3). This contributes to a moderate enhancement of the overall polarization intensity of the material (Fig. 2c4).

(iii) For the ferroelectric phase, strong relative displacement between positive and negative charges leads to large spontaneous polarization and long-range order (Supplementary Note 2.1 and Fig. S4a). Under the influence of the superparaelectric matrix, as shown in Figs. 2d1 and S4b, the local ferroelectric region retains a relatively high polarization magnitude, but its orientation becomes disordered. As a result, without an external field, the net polarization of the overall system does not increase significantly. Influenced by the superparaelectric environment, the polarization in the ferroelectric region can also respond rapidly to an external field (Supplementary Note 2.2 and Fig. S5). As shown in Fig. 2d2, under an applied electric field, the polarization in the superparaelectric matrix responds quickly and significantly facilitates the switching of polarization in the ferro-electric region. Owing to the strongly polar nature of ferroelectric polarization, the whole system exhibits high polarization intensity. Under a reverse electric field, as shown in Fig. 2d3, due to the pro-moting effect of superparaelectric polarization on the response of ferroelectric polarization, the overall polarization of the system can still respond rapidly, showing weak hysteresis. Therefore, the $P–E$ loop of the system is characterized by a substantially increased polarization intensity with nearly unchanged remanent polarization (Fig. 2d4). Since the spontaneous polarization of the ferroelectric phase exceeds

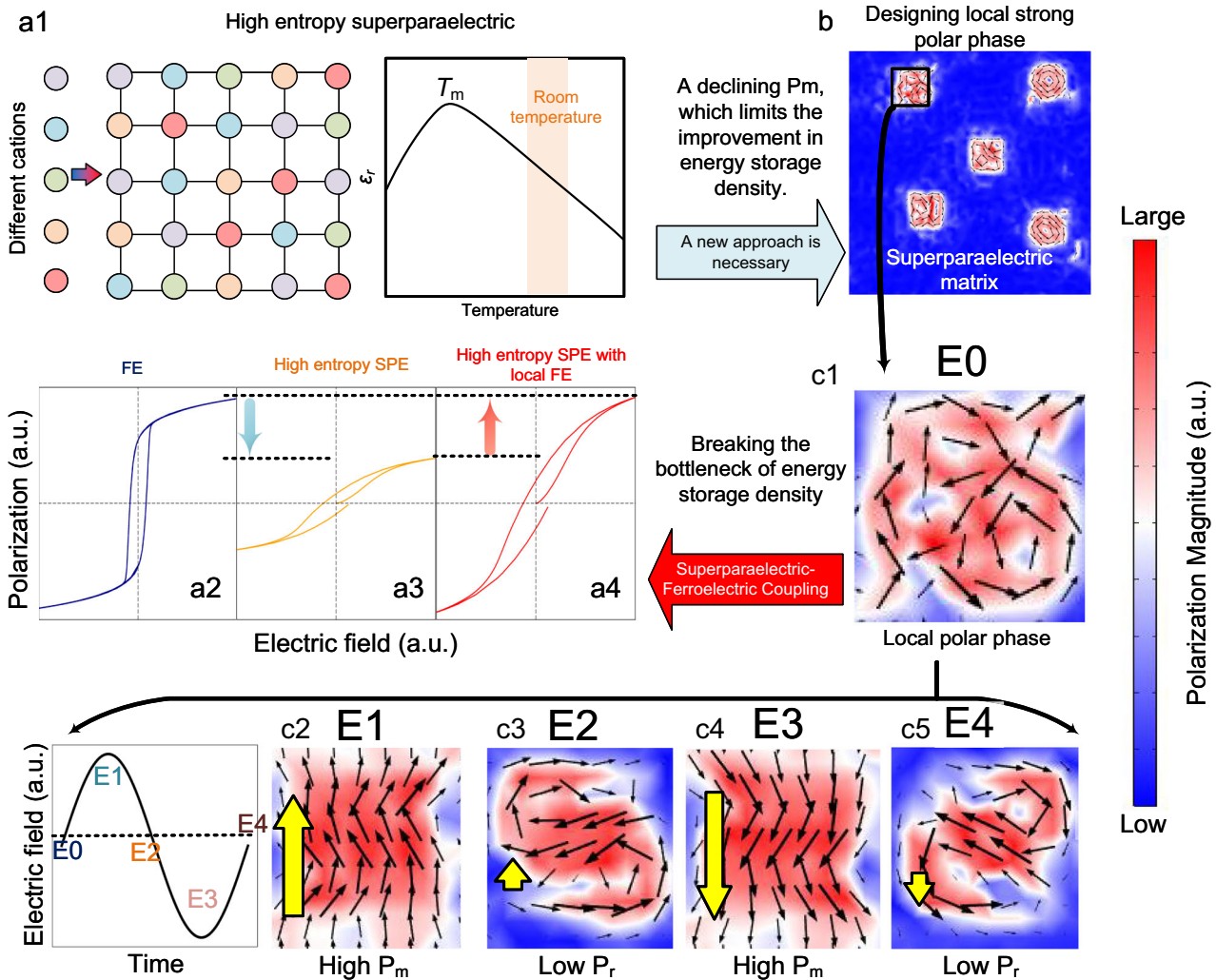

**Fig. 1 | Schematic illustration of the strategy: local ferroelectric–global superparaelectric. a1** Schematic of a high-entropy superparaelectric state. Due to enhanced random fields, the domain structure is refined, presenting a superpara-electric configuration. This reduces losses but also weakens the material's saturation polarization, limiting the improvement in energy storage density. Comparison of P-E hysteresis loops: **a2** Ferroelectric material; **a3** High-entropy superpara-electric material; **a4** High-entropy superparaelectric material with introduced localized ferroelectric phases. **b** Designed polarization structure of the ferroelectric phase within the superparaelectric state, based on phase-field simulations. **c** Polarization structure evolution of the ferroelectric phase within the superpara-electric state, where the colorbar represents the normalized length of polarization, with blue indicating short and red indicating long: **c1** Under the influence of the surrounding superparaelectric polarization, the ferroelectric phase retains a large polarization magnitude but struggles to maintain a uniform polarization direction. **c2** Upon application of a positive electric field **E1**, the superparaelectric polariza-tions rapidly switch upward, significantly promoting the switching of the ferro-electric phase polarization, leading to high saturation polarization. **c3** During the gradual removal of the positive electric field (**E2**), the superparaelectric polariza-tions facilitate the reversal of the ferroelectric phase polarization, resulting in low remanent polarization. **c4 – c5** Application of a negative electric field induces analogous effects on the material's polarization as observed with the positive field.

that of the antiferroelectric phase, the energy storage density of the system can be significantly enhanced.

In summary, the phase-field simulation results indicate that introducing a ferroelectric phase into the superparaelectric matrix can greatly enhance polarization intensity with almost no compromise in efficiency, thereby facilitating the attainment of excellent energy storage density.

**Experimental validation**

To experimentally validate the effectiveness of the aforementioned simulation results, paraelectric $CaTiO_3$, ferroelectric $BaTiO_3$, and $PbTiO_3$, and antiferroelectric $PbZrO_3$ were introduced into a super-paraelectric high-entropy matrix. The structural features and under-lying electronic origins of these components were analyzed using density functional theory calculations (Supplementary Note 3 and Fig. S6). Specifically, structural relaxation was performed to evaluate

the type and degree of structure distortion in each compound, while charge density and partial density of states (PDOS) analyses were employed to uncover the electronic configuration origins of these structural variations. As shown in Fig. 3a, $CaTiO_3$ has a tolerance factor close to 1, exhibits a cubic unit cell with no relative displacement between cations and anions, and is a typical paraelectric phase. As shown in Fig. 3b, $BaTiO_3$ has a tolerance factor of ~1.06, which favors the formation of a ferroelectric tetragonal structure. This distortion is achieved through hybridization between Ti and O (Figs. S7a, b and S8a, b), where Ti possesses unoccupied d orbitals that facilitate orbital mixing with O 2p states, enabling its off-center displacement and resulting in a relative cation-anion shift along the [001] direction. As shown in Fig. 3c, although $PbTiO_3$ has a relatively lower tolerance factor of ~1.02, the presence of the lone pair on the Pb ion (Figs. S7c, d and S8c, d) significantly enhances its ferroelectric distortion, far exceeding that of $BaTiO_3$[28,30,31]. $PbZrO_3$ has a tolerance factor less than

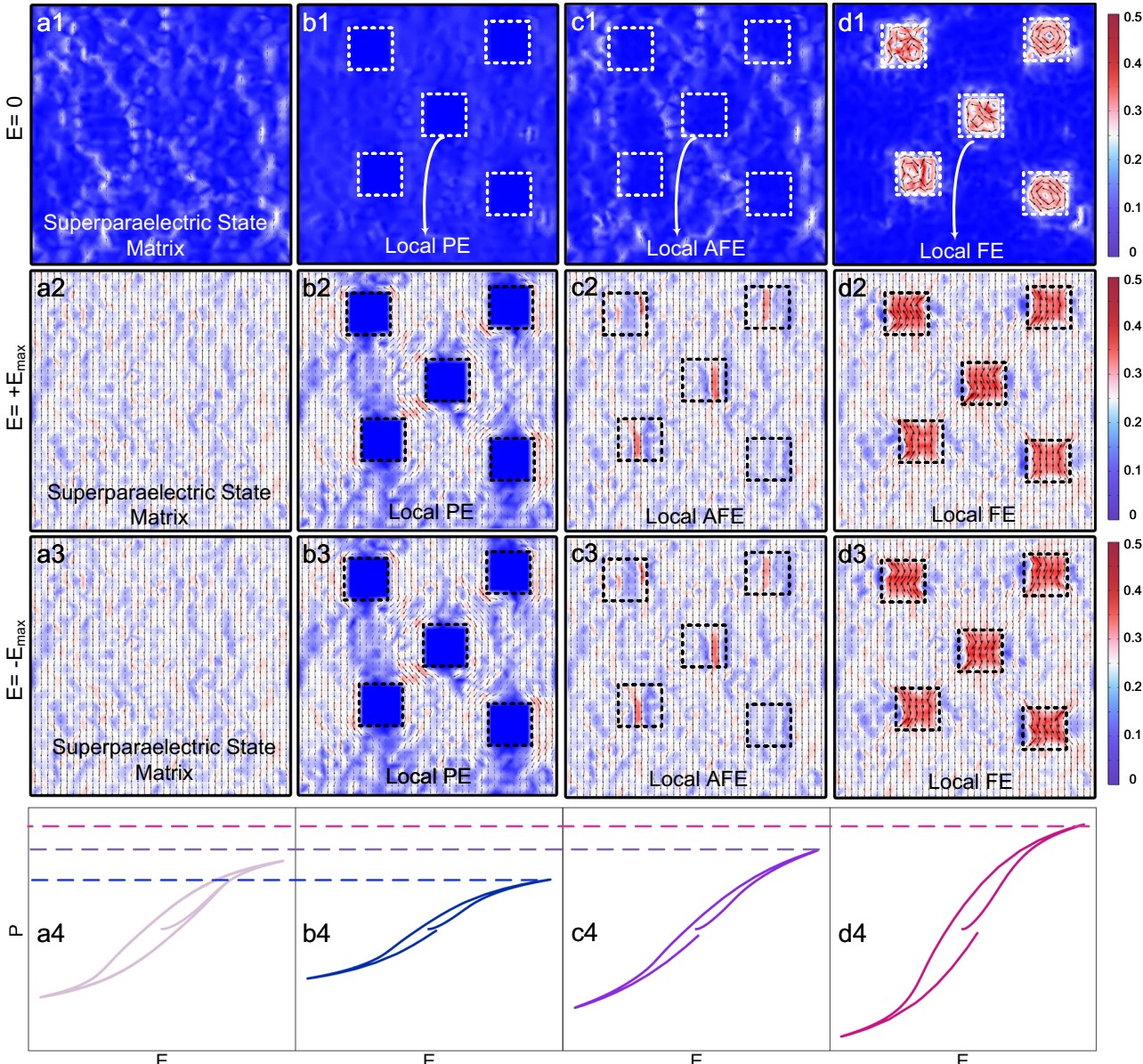

**Fig. 2 | Phase-field simulations of the superparaelectric matrix and the introduction of paraelectric, antiferroelectric, and ferroelectric phases into the superparaelectric state.** Superparaelectric matrix: **a1** polarization structure without external electric field, **a2** under maximum positive external electric field, **a3** under maximum negative external electric field, and the corresponding **a4** schematic of the simulated P-E loop. Introduction of local paraelectric phase into the superparaelectric matrix: **b1** polarization structure without external electric field, **b2** under maximum positive external electric field, **b3** under maximum negative external electric field, and the corresponding **b4** schematic of the simulated P-E loop. Introduction of local antiferroelectric phase into the superparaelectric matrix: **c1** polarization structure without external electric field, **c2** under maximum positive external electric field, **c3** under maximum negative external electric field, and the corresponding **c4** schematic of the simulated P-E loop. Introduction of ferroelectric phase into the superparaelectric matrix: **d1** polarization structure without external electric field, **d2** under maximum positive external electric field, **d3** under maximum negative external electric field, and the corresponding **d4** schematic of the simulated P-E loop. The colorbar represents the normalized length of polarization, with blue indicating short and red indicating long.

1, indicating a distorted crystal structure; simultaneously, both cations −Pb (with its lone-pair electrons) and Zr (with its empty d orbitals)− undergo orbital hybridization with oxygen ions (Figs. S7e, f and S8e, f), contributing to the complex polarization behavior characteristic of antiferroelectrics[32]. As shown in Fig. 3d, these structural features, combined with electronic interactions, lead to the formation of two opposing spontaneous polarizations within the material. However, as these polarizations macroscopically cancel each other out, the net polarization is zero, thus presenting a typical antiferroelectric phase.

We selected BNKLSTZ high-entropy ceramics as the matrix. The superparaelectric state in relaxor ferroelectrics is defined as the temperature range between $T_m$ (the temperature of maximum dielectric permittivity) and $T_B$ (the Burns temperature), where polar nanoregions (PNRs) begin to form but remain dynamically fluctuating with minimized coupling and switching barriers[1,12,33]. As shown in Figs. S9a, b, temperature-related dielectric permittivity indicates that the high-entropy matrix has a $T_m$ (~ −101 °C) far below room temperature and $T_B$ (-100 °C) above room temperature, confirming its room-temperature superparaelectric state. We incorporated 20 mol% of $CaTiO_3$, $BaTiO_3$, $PbTiO_3$, and $PbZrO_3$ into the superparaelectric matrix (abbreviated as 20CT, 20BT, 20PT, and 20PZ, respectively). These high-entropy ceramics possess the configurational entropy of

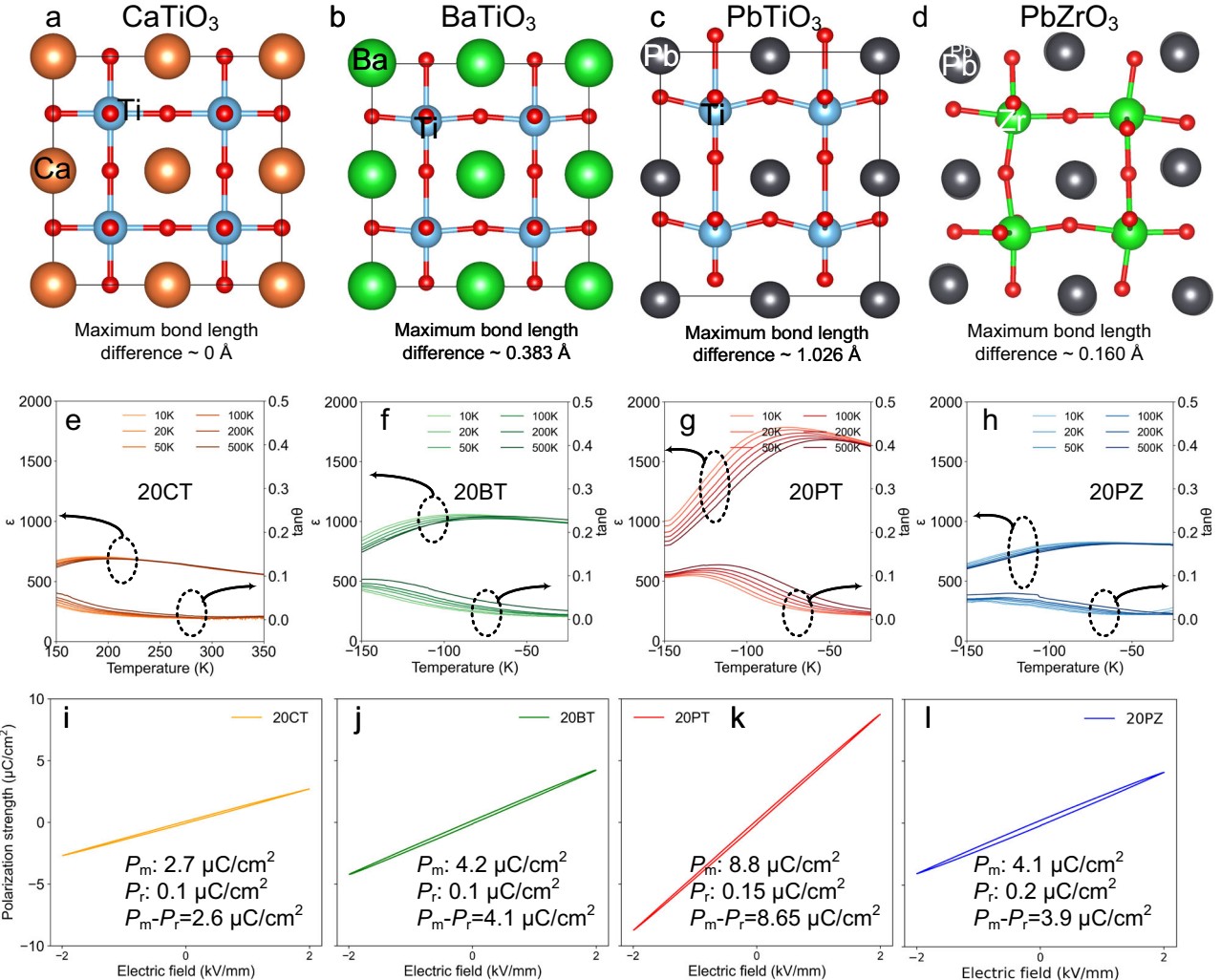

**Fig. 3 | Experimental validation of the influence mechanisms of different ABO₃ components in high-entropy superparaelectrics.** Density functional theory calculation of the 2 × 2 × 2 supercell: **a** paraelectric CaTiO₃, **b** ferroelectric BaTiO₃, **c** strongly ferroelectric PbTiO₃, and **d** antiferroelectric PbZrO₃; Temperature-dependent dielectric constant and loss of **e** 20 mol% CaTiO3, **f** 20 mol% BaTiO₃, **g** 20 mol% PbTiO₃, and **h** 20 mol% PbZrO₃ modified high-entropy BNKLSTZ ceramics. *P–E* hysteresis loops of **i** 20 mol% CaTiO₃, **j** 20 mol% BaTiO₃, **k** 20 mol% PbTiO₃, and **l** 20 mol% PbZrO₃ modified high-entropy BNKLSTZ ceramics.

~1.73–1.88R, placing them within the high-entropy regime ($S_{config} \geq 1.5R$). XRD results (Fig. S10) show that all ceramics exhibit a pure perovskite structure with no impurity peaks, indicating complete solid solubility of all components within the high-entropy matrix. SEM observations (Fig. S11) reveal that these ceramics have dense microstructures with similar grain sizes of ~1 μm. As shown in Figs. 3e–h, after the introduction of these components, the ceramic still maintains a similar $T_m$ below room temperature (~ −92 °C for CT and ~−93 °C for BT, ~−80 °C for PT, and ~−86 °C for PZ @ 10 K Hz), preserving the superparaelectric structure. The relaxation degree γ was obtained by fitting the dielectric temperature spectra with the modified Curie-Weiss law (Fig. S12). For an ideal ferroelectric, γ is 1, while for a relaxor ferroelectric, it is 2. As shown in Fig. S12, 20CT, 20BT, and 20PT ceramics all exhibit high γ values of ~1.8, whereas 20PZ has a lower γ of ~1.5. A γ value close to two indicates that the introduction of ferroelectric phases (BT and PT) does not disrupt the relaxor features essential for the superparaelectric state. Furthermore, the activation energy $E_a$ was obtained by fitting the dielectric spectra with the Vogel-Fulcher law (Fig. S13). $E_a$ reflects the energy required for the response of polar nanoregions (PNRs). The $E_a$ values for the different systems are relatively close (0.02–0.07 eV), suggesting that under an applied electric field, highly polar entities can respond rapidly, leading to high saturation polarization and low remanent polarization.

P-E hysteresis loops measured at 2 kV/mm directly verify the phase-field simulation results. As shown in Figs. 3i–l, the *P–E* curves of all ceramics are slim but exhibit different characteristics depending on the component. Compared to the matrix (Fig. S9c), the maximum polarization of 20CT decreases from 3.0 μC/cm² to 2.7 μC/cm², which is attributed to the lack of polarization in the paraelectric phase under an electric field. The spontaneous polarizations present in 20PZ and 20BT enhance $P_m$ to 4.1 μC/cm² and 4.2 μC/cm², respectively. Notably, the $P_m$ of 20PT increases significantly to 8.8 μC/cm², attributable to the strong polar nature of PT. Furthermore, the remanent polarization is nearly identical for all ceramics. Due to the significantly increased saturation polarization and nearly unchanged remanent polarization, the high-entropy ceramic with PT addition exhibits a markedly improved $\Delta P = P_m − P_r$ (8.65 μC/cm²), which is beneficial for achieving higher energy storage density.

Both the experimental and theoretical calculation results demonstrate that constructing strongly ferroelectric phases within a superparaelectric matrix can significantly enhance saturation polarization while maintaining a low remanent polarization, thereby showing great potential for improving energy storage. It is worth noting that the *P–E* loops shown in Fig. 3i–l were measured at a relatively low electric field of 2 kV/mm. This field strength was deliberately chosen to qualitatively verify the trends predicted by phase-field simulations.

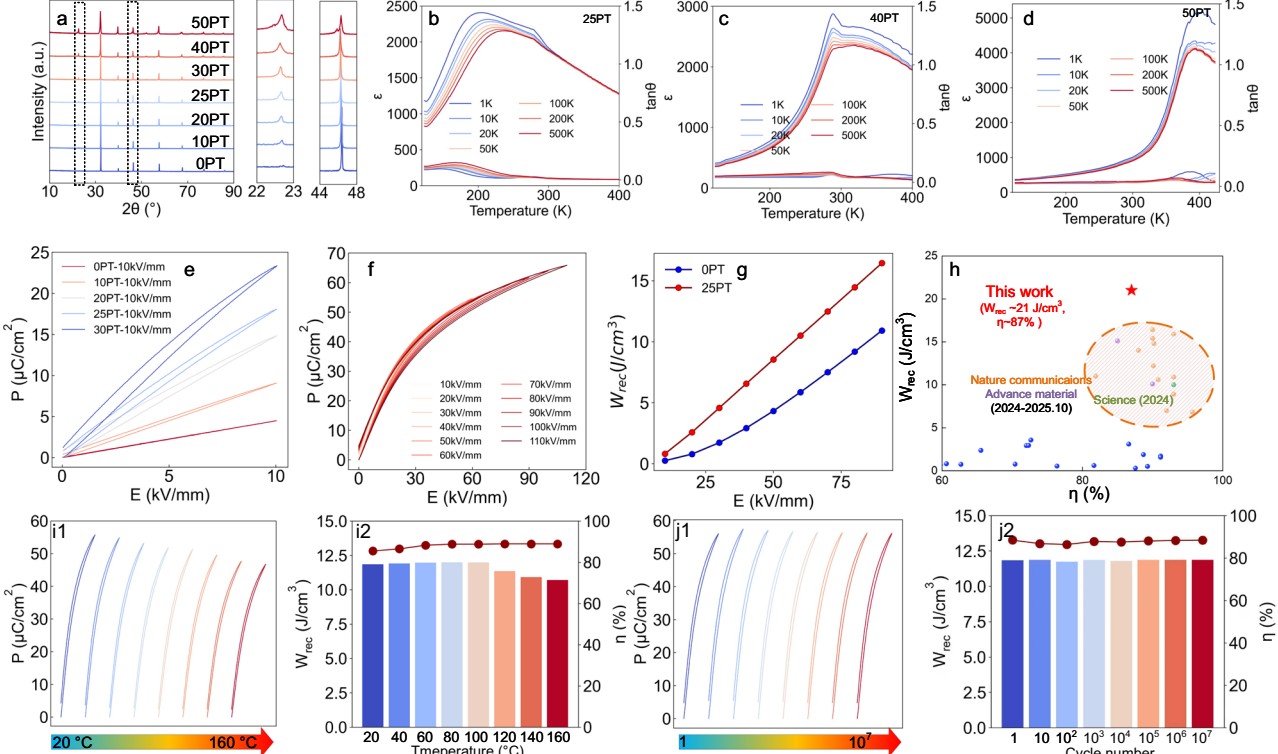

**Fig. 4 | Designing ultrahigh energy storage in high entropy superparaelectric (HESPE). a** XRD patterns of BNKLSTZ-xPT (x = 0, 0.1, 0.2, 0.25, 0.3, 0.4, 0.5) ceramics with different contents of the strong ferroelectric component PT; Temperature-dependent dielectric constant and loss for **b** x = 0.25, **c** x = 0.4, and **d** x = 0.5; **e** Unipolar P–E loops of x = 0 - 0.3 @ 10 kV/mm; **f** Unipolar P–E loops of x = 0.25 under various electric fields; **g** Comparison of the recoverable energy storage density ($W_{rec}$) between x = 0 and x = 0.25; **h** Comparison of $W_{rec}$ and $\eta$ between x = 0.25 and other reported works; **i** P-E hysteresis loops of the BNKLSTZ-25PT ceramic at 70 kV/mm over a wide range of **i1** the temperature from 20 to 160 °C with **i2** the corresponding $W_{rec}$ and $\eta$; **j** P–E hysteresis loops of the BNKLSTZ-25PT ceramic at 70 kV/mm over a wide range of **j1** cycle number from 1 to $10^7$ with **j2** the corresponding $W_{rec}$ and $\eta$.

The results obtained under this low field consistently support the simulation conclusions. Building on this mechanistic understanding, subsequent energy storage performance optimization by designing PT content was conducted under breakdown field

## Designing ultrahigh energy storage density

The aforementioned results indicate that, at the same content, the incorporation of the ferroelectric component PbTiO₃ can significantly increase $P_m$ without sacrificing $P_r$, thereby facilitating the enhancement of energy storage density, and its effect far exceeds that of the weak ferroelectric component BaTiO₃. To optimize energy storage performance and further analyze the role of PbTiO₃, we introduced 0, 10, 20, 25, 30, 40, and 50 mol% PT (abbreviated as 0PT, 10PT, 20PT, 25PT, 30PT, 40PT, 50PT) into the superparaelectric matrix of BNKLSTZ, and conducted structural characterization and performance testing. As shown in Fig. 4a, all ceramics exhibit a pure perovskite structure without impurity peaks, indicating that all PT has been incorporated into the high-entropy matrix. The (100) diffraction peak at ~22.5° in XRD is related to the polarity of the ceramics, with a higher (100) peak intensity indicating enhanced ceramic polarity. As the PT content increases, the (100) diffraction peak gradually appears, and its relative intensity increases, demonstrating that the introduction of PT significantly enhances the polarity of the matrix. Furthermore, when the PT doping content is ≤30 mol%, the diffraction peaks remain single peaks, indicating that the average phase structure of the material is still dominated by the pseudo-cubic phase. When the PT content increases to 40 mol%, the main peak at ~32° splits, exhibiting a peak shape with lower left and higher right sides, suggesting a significant enhancement in ferroelectric distortion. As the PT content further increases to 50%, in addition to the main peak splitting, the characteristic peaks between

44 and 48° also show clear splitting, indicating a further increase in T-phase ferroelectric distortion. The emergence of this highly polar T-phase can be attributed to the gradual expansion of the influence of PbTiO₃ from local to the entire system. To investigate the changes in phase content when the PT doping content is ≤30 mol%, as shown in Fig. S14, we performed XRD refinement. The results indicate that the introduction of PT increases the T-phase content in the matrix (from ~18% to ~40%), although the overall tetragonal distortion remains weak (c/a ~1.005). SEM (Fig. S15) shows that all ceramics have a dense and fine-grained structure (grain size ~1 μm), so the changes in performance are primarily attributed to alterations in their polarization structure.

The temperature-related dielectric permittivity and loss results (Figs. 4b–d and S16a–d) are consistent with XRD. When the PT content is ≤30 mol%, the $T_m$ peak is below room temperature, and $T_B$ is above room temperature, indicating that the ceramics are in a superparaelectric state (Figs. S16e–h). The presence of frequency dispersion and the appearance of low-temperature dielectric loss peaks both suggest a large number of PNRs within the ceramics. As shown in Figs. 4c, d, when the PT content further increases to 40–50 mol%, the $T_m$ peak shifts above room temperature, and both frequency dispersion and low-temperature loss peaks disappear, indicating that excessive PT disrupts the room-temperature superparaelectric structure. These observations are consistent with the XRD results mentioned above, suggesting that the influence of PbTiO₃ has expanded from local to global, transforming the material from a superparaelectric matrix to a strong ferroelectric PbTiO₃-dominated matrix. This transformation of the matrix is also reflected in its hysteresis loops and piezoelectric performances. As shown in Fig. S17, 40PT and 50PT ceramics exhibit typical ferroelectric P–E curves and moderate piezoelectric

coefficients (20 and 100 pC/N, respectively). For energy storage characteristics, slim $P$–$E$ loops with high $P_m$ and low $P_r$ are desired. Therefore, in the following analysis of dielectric and energy storage performance, we focus primarily on ceramics with PT content ≤30%. As shown in Figs. S18–S19, dielectric temperature spectra of ceramics with 0–30% PT content were fitted using the modified Curie-Weiss law and the Vogel-Fulcher law[34]. Within this range, the ceramics maintain a high degree of relaxation ($\gamma \sim 1.8$) and low activation energy ($E_a \sim 0.02$ eV), which is favorable for energy storage applications. Unipolar P-E loops of ceramics with 0–30 mol% PT doping were further tested, showing that the polarization intensity of the ceramics significantly increases with introducing PT, indicating the successful implementation of our strategy (Fig. 3e). In particular, the 25PT ceramic exhibits the most energy storage performance, achieving an ultrahigh energy storage density of 21 J/cm³ and an energy storage efficiency of 87% under a breakdown field strength of 110 kV/mm (Fig. 4f). To verify the statistical reliability of the reported energy storage performance, we performed Weibull analysis on the breakdown strength of the 25PT ceramic. As shown in Fig. S20a, the breakdown data exhibit excellent linear fitting to the Weibull distribution, yielding a characteristic breakdown strength of $E_b \sim 111.5$ kV/mm with a high shape parameter of $\beta \sim 19.4$. We further evaluated the sample-to-sample reproducibility of the polarization response. As shown in Fig. S20b, four independently prepared 25PT samples exhibit nearly identical $P_m$ and $P_r$ values as a function of electric field. Under the characteristic breakdown field of 110 kV/mm, these samples yield consistent energy storage densities of $21 \pm 1$ J/cm³ and efficiencies of $87 \pm 1\%$. These results demonstrate that the ultrahigh energy storage performance of the 25PT ceramic is both reproducible and reliable.

Figure 4g compares the energy storage densities of 0PT and 25PT ceramics under the same field strength, showing that the significant increase in polarization intensity due to the introduction of the ferroelectric component greatly enhances the energy storage density. As shown in Fig. 4h, comparing the performance of 25PT with other currently reported ceramics, it is evident that 25PT falls within the highest performance range reported to date, further demonstrating the effectiveness of this strategy[14,15,20,23,35–39]. For practical applications, the stability of energy storage performance under varying thermal and electrical conditions is equally important. As shown in Fig. 4i1–i2, the 25PT ceramic exhibits excellent temperature stability in the range of 20–160 °C at 70 kV/mm, with $W_{rec} \sim 11.3 \pm 0.6$ J/cm³ and $\eta \sim 87.2 \pm 1.7\%$, demonstrating minimal degradation even at elevated temperatures. The fatigue test (Fig. 4j1–j2) reveals exceptional cycling reliability, with $W_{rec}$ and $\eta$ exhibiting negligible variation after $10^7$ charge-discharge cycles ($W_{rec} \sim 11.8 \pm 0.1$ J/cm³, $\eta \sim 87.9 \pm 0.8\%$). Furthermore, the frequency-dependent measurements (Fig. S21) show that $W_{rec}$ and $\eta$ remain stable across a wide frequency range of 1–300 Hz ($W_{rec} \sim 11.5 \pm 0.5$ J/cm³, $\eta \sim 87.1 \pm 2.1\%$), indicating rapid polarization response capability. As shown in Fig. S22, charge-discharge tests on 25PT indicate that, under a field strength of 46 kV/mm, 25PT possesses a high discharge density ($W_d \sim 6.7$ J/cm³) and ultrafast discharge speed ($t_{0.9} \sim 37$ ns), along with a high current density ($C_D \sim 2344$ A/cm²) and power density ($P_D \sim 539$ MW/cm³), showing great potential for applications.

## Multi-scale characterizations for *0PT* and *25PT* ceramics
To investigate the effect of 25PT incorporation on the microstructure of the superparaelectric matrix, we conducted multi-scale structural characterization on 0PT and 25PT ceramic samples. As shown in Fig. 5a, b, in-situ Raman spectra from −150 to 200 °C reveal that 25PT exhibits stronger Raman peak intensity in the low wavenumber range (50–300 cm⁻¹) across a wide temperature range, while in the medium wavenumber range (400–1000 cm⁻¹), the Raman spectra of the two ceramics are relatively similar. Figure 5c compares the peaks of 0PT and 25PT ceramics in the 100–150 cm⁻¹ wavenumber range, showing

that 25PT has a stronger Raman peak. For $ABO_3$ ferroelectric materials, this range is related to A-O bond vibrations[40], and the stronger peak intensity is associated with the introduction of Pb at the A-site. The lone pair electrons of Pb lead to hybridization with oxygen, reducing crystal symmetry in the originally superparaelectric matrix (where the A-site vibration Raman peak signal is weak), thereby making these originally Raman-inactive rotational modes become Raman-active and enhanced. Furthermore, as shown in Fig. 5d, the peaks in the 200–400 cm⁻¹ region also show significant enhancement. This region corresponds to the bending modes of $BO_6$ octahedra and vibrations related to B-site ion displacements[40], so the enhancement of peaks indicates that the lone pair electron effect of $Pb^{2+}$ can propagate through the lattice, driving B-site ions away from the center of their oxygen octahedra, which favors enhanced polarization length. The peaks in the 400–600 cm⁻¹ region mainly correspond to stretching vibrations of B-O bonds, reflecting the strength and bond length of B-O bonds[41]. As shown in Fig. S23, the peak intensities of the two ceramics in the 400–600 cm⁻¹ range do not change significantly, indicating that the incorporation of 25PT primarily affects the geometric configuration of the lattice (angles, ion displacements) without drastically altering the covalent bond strength of the B-O bonds themselves. Figure S24 compares the XRD patterns of the two ceramics in the 25–200 °C range, showing that they possess similar pseudo-cubic average phase structures. Combining in-situ Raman spectroscopy and XRD results, it is evident that the introduction of 25PT significantly enhances the local ferroelectric structure without affecting the average phase structure (superparaelectric structure).

Transmission electron microscopy and atomic-scale STEM observations provide more direct evidence. As shown in Fig. S25a, for the superparaelectric high-entropy matrix, no obvious domain structure is observed at low magnification, which is due to the disruption of domain structures by high entropy effects. In contrast, nanodomain structures can be observed in some local regions of the 25PT ceramic (Fig. S25b), indicating that the introduction of PT greatly enhances the local polarization of the material. As shown in Fig. 5e, in STEM observations, the atomic contrast is proportional to the square of the atomic number, allowing chemical distribution analysis through ion contrast. Furthermore, for $ABO_3$ materials, under the [110] zone axis, the relative displacement of B-site ions from the center of the nearest A-site and O atoms reflects the polarization direction, and the magnitude of the relative displacement indicates the polarization intensity[22,42]. We extracted the A-site ion contrast of 0PT and 25PT to obtain Fig. 5f1–f2, showing that the A-site ion contrast of 25PT is significantly higher than that of 0PT ceramics, which is due to the larger atomic number of Pb, resulting in brighter contrast. Additionally, in 25PT, the distribution of A-site ions shows obvious inhomogeneity, indicating that Pb exhibits typical localization in the matrix. Figure 5g1–g2 compares the polarization configurations of 0PT and 25PT. Both ceramics exhibit fine polar nanoregion structures with diverse polarization directions, showing nanoscale coexistence of R-O-T-C multiphases. However, it is worth noting that the polarization length of 25PT is significantly enhanced. Combining chemical distribution and polarization configuration analyses, the localized distribution of Pb allows the overall structure to remain in a superparaelectric matrix (with PNR sizes still fine, ~1–2 nm), while the local polarization length is significantly enhanced. As indicated by the aforementioned phase-field simulation results (Fig. 2d1), such a structure is conducive to achieving high saturation polarization and low remanent polarization, resulting in enhanced energy storage density.

## Lone pair effect: from local to global, from energy storage to piezoelectricity
Based on the above analysis, compared to the $BaTiO_3$, the introduction of the strongly ferroelectric component $PbTiO_3$ can significantly enhance the energy storage properties of high entropy

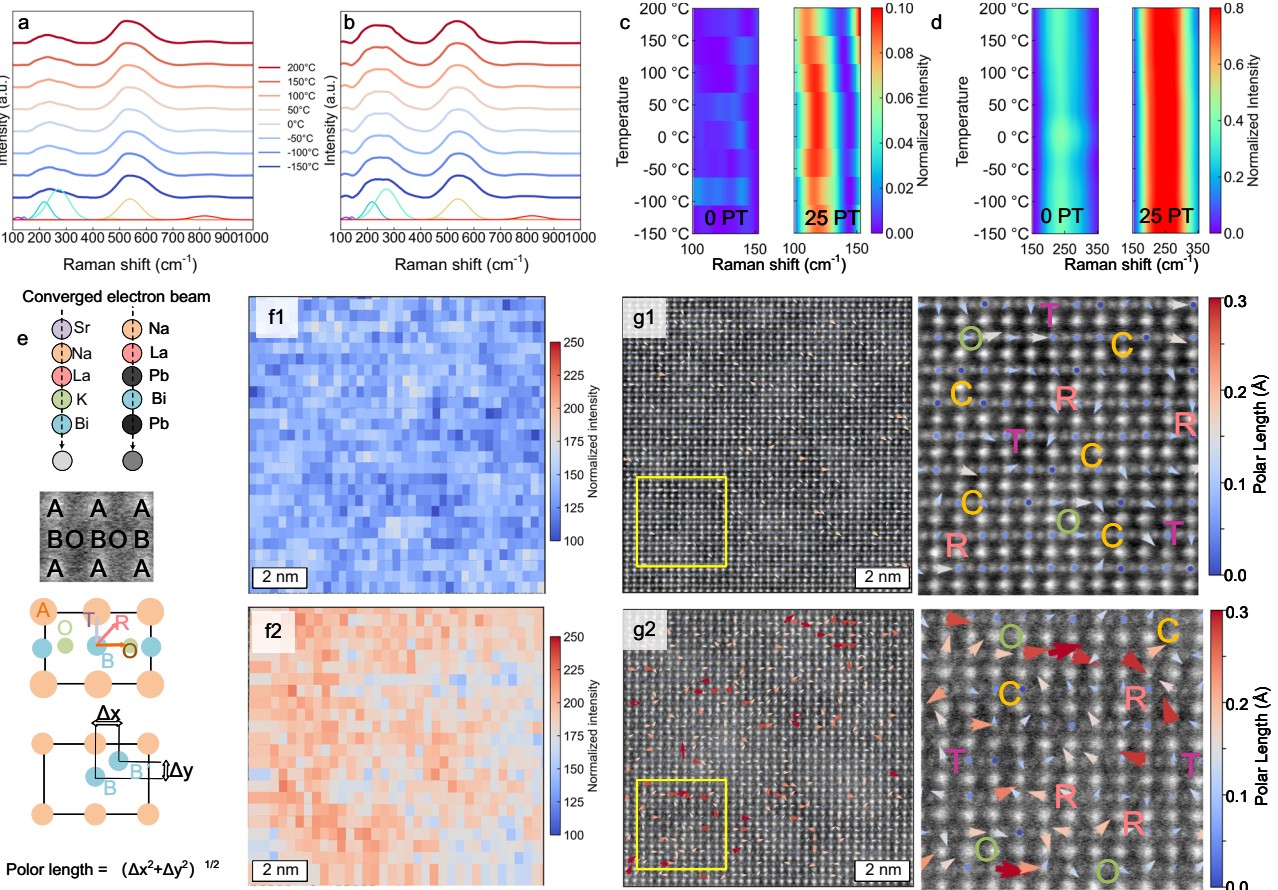

**Fig. 5 | Multiscale characterizations of BNKLSTZ-xPT ($x = 0$ and $x = 0.25$).** In-situ temperature-dependent Raman spectra of **a** $x = 0$ and **b** $x = 0.25$; Comparison of Raman peaks between $x = 0$ and $x = 0.25$ in **c** the low-wavenumber region (100-150 cm$^{-1}$) and **d** the mid-wavenumber region (150–350 cm$^{-1}$); **e** Schematic illustration of atomic-scale STEM observation along the [110] zone axis; A-site ion contrast mapping for (**f1**) $x = 0$ and (**f2**) $x = 0.25$; Polarization direction and polarization length mapping for (**g1**) $x = 0$ and (**g2**) $x = 0.25$.

superparaelectric (HESPE). We also compared the $P$–$E$ loops and small-signal d$_{33}$ measurements for BNKLSTZ with the introduction of 40BaTiO$_3$ and 40PbTiO$_3$, respectively (Fig. S26). It can be seen that the former still maintains slim $P$–$E$ hysteresis loops and no measurable piezoelectric coefficient. In contrast, the latter exhibits enhanced ferroelectricity and a slight piezoelectric response (~20 pC/N). It is particularly noteworthy that both components contain the same Ti content. Therefore, the enhanced polarity within the high entropy SPE state can be primarily attributed to the role of the strongly polar element Pb. Furthermore, as the PT content increases further to 50%, a moderate piezoelectric coefficient (100 pC/N) emerges. These observed trends in structural and property evolution indicate that the influence of highly polar Pb shifts from localized to global effects on the superparaelectric state, thereby transforming the material's functionality from energy storage to piezoelectricity. It is particularly noteworthy that most high-entropy ferroelectrics reported to date are primarily targeted for energy storage applications. Therefore, understanding the structural transition mechanism described above is expected to provide insights for broader ferroelectric applications in the future.

Here, we conduct a further mechanistic investigation by combining TEM, atomic-scale STEM, and DFT calculations. As shown in Fig. S27, TEM observations of 25PT, 40PT, and 50PT reveal that with the introduction of more strongly polar Pb, an increasing number of lamellar ferroelectric domain structures (domain size also increases from ~20 nm to ~50 nm) appear in the samples, confirming that Pb can significantly strengthen the ferroelectric polar. To analyze the

mechanism of the strongly polar element Pb, we performed STEM measurements on 25PT, 40PT, and 50PT along the [100] zone axis. As shown in Fig. 6a, the distribution of A-site elements can be analyzed by extracting their ionic contrast. For the heavier Pb element, it results in a brighter contrast for the A-site ions. Furthermore, along the [100] zone axis, the polarization direction and magnitude can be analyzed via the displacement of B-site ions relative to the center of the surrounding A-site sublattice[22,30,31,43,44]. As shown in Figs. 6b and S28, in 25PT, the B-site ions are clearly visible, and the brightness difference between A- and B-site ions is not significant. With increasing Pb content, as shown in Fig. 6c, d, the brightness of A-site ions progressively increases relative to that of B-site ions, while the B-site ion intensity gradually darkens. This is consistent with the previous discussion: the introduction of the heavy element Pb markedly enhances the A-site brightness, leading to an increasing contrast difference between the A- and B-site elements. We further extracted the contrast distribution of A-site ions, as shown in Fig. 6e–g. In 25PT, due to the relatively low Pb content (25%) and the significant proportion of other elements, the ionic contrast shows obvious brightness inhomogeneity, indicating a relatively non-uniform Pb distribution at this stage (Fig. 6e). As the PT content increases (Fig. 6f, g), the average ionic contrast of the A-site becomes brighter and its distribution becomes more uniform, suggesting that the introduction of more Pb leads to increasingly homogeneous distribution. Influenced by the Pb content and distribution, the polarization configuration of 25PT, shown in Figs. 6h and S29a, remains similar to that of 0PT (Fig. S30). A significant amount of non-polar phase persists, with strongly polar nanoregions (size 1–2 nm)

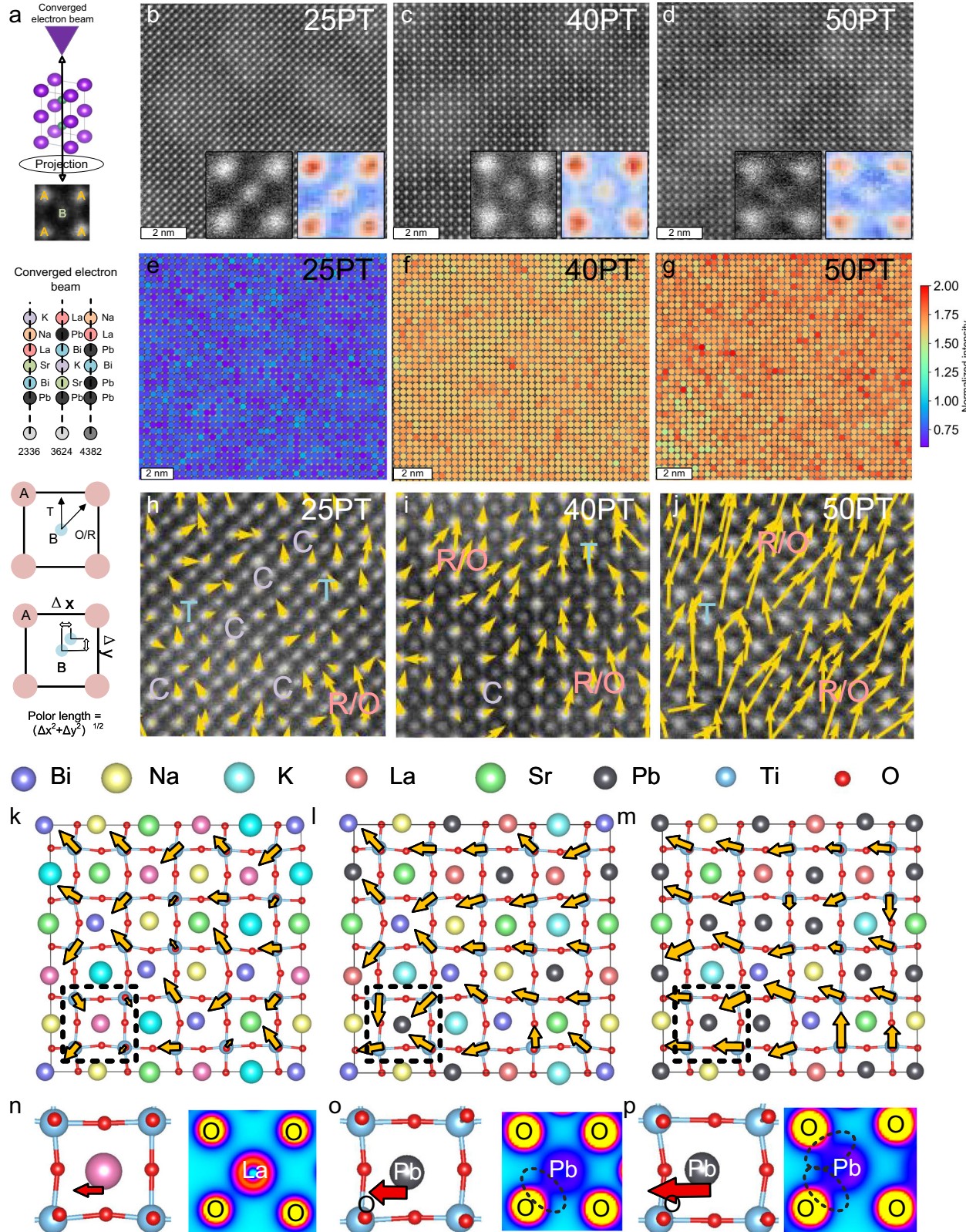

**Fig. 6 | Mechanism analysis of how strongly polar element Pb enhances long-range ferroelectric order in a superparaelectric matrix. a** Schematic illustration of [100]-zone-axis HAADF observation; atomic-scale HAADF images of **b** 25PT, **c** 40PT, and **d** 50PT; A-site contrast distribution of **e** 25PT, **f** 40PT, and **g** 50PT; Polarization configurations of **h** 25PT, **i** 40PT, and **j** 50PT; Crystal structures from density functional theory calculations for **k** BNKLST, **l** 20% Pb-substituted BNKLST, and **m** 48% Pb-substituted BNKLST; Ferroelectric distortion and charge density distribution at **n** the weakly polar La site, **o** the Pb site in the 20% Pb-substituted structure, and **p** the Pb site in the 48% Pb-substituted structure.

embedded within the non-polar matrix. Consistent with observations along the [110] zone axis (Fig. 5g2), the polarization configuration in 25PT is highly flexible, exhibiting a local coexistence of R/O-T-C multiphase structures, which provides the prerequisite for excellent energy storage performance. When the PT content increases to 40%, as shown in Figs. 6i and S29b, the volume fraction of the polar phase increases, and the polarization magnitude is enhanced. As shown in Figs. 6j and S29c, when the PT content is further increased to 50 mol%, the non-polar phase disappears, and the polarization evolves into a long-range ordered structure.

To analyze the mechanism by which Pb affects the crystal structure, we performed DFT calculations (Supplementary Note 4 and Fig. S31). As shown in Fig. S31a, we constructed a $5\times5\times1$ initial supercell with the C-phase, where Bi, Na, K, La, and Sr were randomly distributed to simulate the high-entropy matrix. After structural relaxation, as shown in Fig. 6k, due to differences in ionic radii and electronic configurations among the A-site elements, the unit cell exhibits severe lattice distortion, disordered polarization orientations, and non-uniform polarization magnitudes. These results align with the current theoretical understanding of the high-entropy strategy: multiple elements occupying the same lattice site enhance random fields (displacement and electric fields), leading to refined polar nanoregions. To simulate the effect of different Pb contents on the supercell structure, we replaced 5, 10, and 12 A-site atoms in the structurally relaxed high-entropy BNKLST cell with Pb (Fig. S31b–d) and performed further structural relaxation. The results show that, as illustrated in Figs. 6l, m and S32, the lattice distortion is significantly enhanced, the polarization magnitude increases, and the disordered local polarization orientations become unified, tending towards the [010] spontaneous polarization direction of the T-phase. Selecting the boxed regions in Fig. 6k–m yields Fig. 6n–p. It can be observed that compared to the weakly polar element La, the ferroelectric distortion associated with Pb is significantly greater. This enhancement becomes more pronounced with increasing Pb content, manifested as a substantial increase in the difference of A-O bond lengths, growing from -0.376 Å for the matrix element La to 0.855 Å for 25PT and 0.929 Å for 50PT. Further charge density calculations for these structures indicate that, as shown in Figs. 6n–p and S33, during this polarity enhancement process, the overlap of electron clouds between Pb and O increases significantly, which can be attributed to the unique lone-pair electron structure of Pb. The hybridization of electron clouds can effectively counteract the increased strain energy resulting from ferroelectric distortion[45]. This progressive enhancement in A-O bond length asymmetry and orbital hybridization with increasing Pb content directly correlates with the microstructural evolution observed in STEM (Fig. 6h–j). At low Pb content (25%Pb), the A-O bond length difference and Pb-O hybridization remain spatially confined within isolated polar nanoregions, preserving the global superparaelectric matrix and resulting in enhanced energy storage performance. At high Pb content, the A-O bond length difference and Pb-O hybridization further increase, indicating that strongly distorted Pb-O bonding networks percolate throughout the entire lattice, leading to long-range ferroelectric order and the emergence of macroscopic piezoelectricity.

Based on the mechanistic insights obtained in these results, further enhancement of energy storage performance can be anticipated by optimizing the distribution of the ferroelectric polarization to maximize the polarization difference ($P_m - P_r$), and by engineering the microstructure to increase the breakdown strength $E_b$ (Supplementary Note 5 and Figs. S34–S35). These strategies, guided by the "local ferroelectric–global superparaelectric" design principle, offer a promising pathway toward even higher energy storage performances in high-entropy ferroelectric ceramics

In summary, this study proposes a strategy for designing localized ferroelectric polarization within high-entropy superparaelectric to modulate energy storage performance. To validate this strategy, the strongly polar $PbTiO_3$ phase was introduced into BNKLSTZ ceramics, achieving an ultrahigh energy storage density of ~21 J/cm³ and an efficiency of ~87%. Moreover, by modulating the distribution of the ferroelectric polarization within a high-entropy superparaelectric matrix, we achieve on-demand enhancement of polarization response, enabling optimization of energy storage and piezoelectric performance in the same material system. Multiscale structural characterization and density functional theory calculations confirm that the lone pair effect of Pb enhances ferroelectric polarization within the high-entropy superparaelectric matrix. When the $PbTiO_3$ content is ≤30%, this effect is confined to localized regions. Macroscopically, the material maintains a superparaelectric structure, while microscopically, it exhibits a nanoscale multiphase coexistence of R-O-T-C structures accompanied by enhanced polarization length. This leads to a flexible polarization response to external fields, resulting in energy storage performance. With a further increase in $PbTiO_3$ content, this effect extends throughout the entire matrix, leading to the emergence of submicro-scale ferroelectric domain structures and enhanced piezoelectric properties. This work not only provides a perspective for future research on the energy storage performance of high-entropy superparaelectrics but also holds potential for inspiring other applications of high-entropy ferroelectrics, such as electrostriction and piezoelectric performance.

## Methods

### Ceramics preparation

The $0.8Bi_{0.2}Na_{0.2}K_{0.2}La_{0.2}Sr_{0.2}Ti_{0.9}Zr_{0.1}O_3$-0.2x ($x = CaTiO_3$, $BaTiO_3$, $PbTiO_3$ and $PbZrO_3$) and $1-xBi_{0.2}Na_{0.2}K_{0.2}La_{0.2}Sr_{0.2}Ti_{0.9}Zr_{0.1}O_3$-$xPbTiO_3$(BNKLSTZ-xPT, $x = 0.0$, 0.10, 0.20, 0.25, 0.30, 0.40 and 0.50) ceramics were prepared by raw powders of $K_2CO_3$(99.0%), $Na_2CO_3$(99.8%), $SrCO_3$(99.0%), $Bi_2O_3$(99.0%), $TiO_2$(98.0%), $La_2O_3$(99.99%), $PbZrO_3$(99.7%), $PbTiO_3$(99.9%), $CaTiO_3$(99.0%), $BaTiO_3$(99.99%) and $ZrO_2$(99.99%) through a conventional solid-state reaction. The raw powders were weighed stoichiometrically and then mixed thoroughly using ball-milling with zirconia balls and absolute ethyl alcohol in a polyethylene container for 12 h. After being calcined at 1100 °C for 2 h, the mixture was ball-milled again for 8 h. The dried mixture was pressed into pellets with a diameter of 10 mm under 200 MPa for 30 s. At last, BNKLSTZ based ceramics were sintered at 1200–1300 °C in air for 2 h.

### Electrical performance testing

The samples are polished into ~0.05 – 0.07 mm in thickness, then coated with gold electrode for *P-E* hysteresis loop measurements. Diamond polishing liquid with particle sizes of 3, 2, and 0.5 microns was used for polishing to ensure low surface roughness. Underdamped and over-damped charge/discharge performance is tested by a capacitor charge-discharge system (CFD-003, Gogo Instruments, China). As shown in Fig. 22a, during the testing process, the left switch is first turned to the left to initiate charging, with a charging time of 500 ns. Subsequently, after charging, the bidirectional switch is turned to the right, allowing the ceramic capacitor to discharge into a load resistor ($R$) of 300 Ω, while the discharge signal is measured over a duration of 500 ns.

### Structural characterizations

The crystal structures of samples were analyzed using in-situ X-ray diffraction (XRD, SMARTLAB, Japan) with Cu Kα-0.1541 nm. The in-situ Raman spectra were obtained by Raman spectrometer (Horiba LabRAM HR Evolution). Surface morphologies of the samples were examined by a field emission scanning electron microscope (Gemini SEM 500, Carl Zeiss, Germany). The dielectric properties of the samples were measured using an LCR meter (E4980A, Agilent). The *P–E* hysteresis loops under different electric fields were measured by Precision Premier II from Radiant Technologies, connected to a high

voltage amplifier. The focused ion beam (FIB) milling technology was used in this work to prepare the TEM sample. A spherical aberration-corrected transmission electron microscope (Thermo Fisher Spectra 300, USA) was used to acquire the bright field observations and high-angle annular dark-field scanning-transmission electron microscopic (HAADF-STEM) images. Based on the HAADF-STEM images, the polarization vector and atom column intensity were calculated and extracted by the MATLAB scripts[46].

**Phase-field simulations**
The detailed methodology for phase-field simulations is presented in Supplementary Notes 1-2.

**DFT calculations**
The detailed methodology for phase-field simulations is presented in Supplementary Note 3-4.

## Data availability
The authors declare that the data supporting the findings of this study are available within the paper and its Supplementary Information files.

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

## Acknowledgements

This work was supported by Fundamental and Interdisciplinary Disciplines Breakthrough plan of the Ministry of Education of China (Grant No. JYB2025XDXM409), the National Natural Science Foundation of China (Grant No. 52532004), the Postdoctoral Fellowship Program and China Postdoctoral Science Foundation under Grant Number BX20250300 and the Fundamental Research Funds for the Central Universities of Central South University. The project was also supported by State Key Laboratory of Powder Metallurgy, Central South University, Changsha, China.

## Author contributions

J.Z., S.Z., and D.Z. conceived and designed the project. T.W. prepared the samples and performed the property measurements. T.W. and J.Z. conducted the DFT calculations and phase-field simulations. M.S and K.Z. performed the TEM and STEM observations and analysed the corresponding data. D.Z., C.K.Z., Z.Y., Z.Z., and X.Z. supervised the experiments. T.W. wrote the manuscript. J.Z. revised the manuscript. All authors discussed the results and contributed to the final manuscript.

## Competing interests

The authors declare no competing interests.
