## [Transparent Peer Review file · Nature Communications]

Enhanced Energy Storage in High-Entropy Superparaelectrics via Local Ferroelectric Polarization

Corresponding Author: Professor Dou Zhang

Version 0:

Reviewer comments:

Reviewer #1

(Remarks to the Author)

This paper proposes an innovative design strategy of "local ferroelectric–global superparaelectric." By introducing the strong ferroelectric phase PbTiO_3 into a high-entropy superparaelectric matrix (BNKLSTZ) and leveraging the coupling between them, an extremely high polarization response under an electric field is achieved, resulting in an ultrahigh recoverable energy storage density ($\sim 21 \text{ J/cm}^3$) and high efficiency ($\sim 87\%$). The research comprehensively employs phase-field simulations, experimental fabrication, multi-scale structural characterization (in-situ XRD, in-situ Raman, atomic-scale STEM, etc.), and first-principles calculations to systematically demonstrate the critical role of the local Pb^{2+} lone pair effect in enhancing local ferroelectric polarization while maintaining the overall superparaelectric structure. The work is systematic and complete, with substantial data, offering significant theoretical and practical guidance for the design of high-performance energy storage ceramics and high-entropy ferroelectric materials. Overall, this is a high-quality study, and I recommend modifications and improvements based on the following specific comments before acceptance.

Specific comments are as follows:

1. Abstract and Introduction:

- (a) The abstract could be further refined to explicitly highlight the core advantage of the "local ferroelectric–global superparaelectric" strategy over a pure high-entropy strategy (i.e., addressing the bottleneck of limited polarization (P_m) in high-entropy systems).
- (b) When reviewing how the high-entropy strategy enhances energy storage performance in the Introduction, a more specific explanation of "how the high-entropy effect leads to the formation of polar nanoregions (PNRs)" would clarify the logical chain.
- (c) The article lacks citations to relevant studies that could support the statements provided related to Fig. 1; for example, "Density functional theory analysis reveals that the presence of the lone pair electrons on Pb is responsible for generating the localized strong polarity, providing the atomic-scale mechanism for the performance enhancement." etc. The authors are asked to cite the original related work that supports your statements.

2. Results and Discussion:

- (a) Phase-field simulations (Fig. 2): The label "P-E loop" in Fig. 2d4 should be clearly stated as "schematic of the simulated P-E loop characteristics."
- (b) Experimental verification (Fig. 3): The test electric field (2 kV/mm) for the P-E loops of different compositions in Fig. 3i-l is relatively low, serving to verify trends. It is recommended to add an explanation in the main text that differences observed under this field are sufficient to support the simulation conclusions, while the subsequently optimized performance is obtained under high breakdown field strengths.
- (c) Verification of SPE state: The authors selected BNKLSTZ high-entropy ceramics as the matrix. As shown in Fig. S9a, temperature-related dielectric permittivity indicate that the high-entropy matrix has a T_m far below room temperature ($\sim -101^\circ\text{C}$), confirming its room-temperature superparaelectric state. The authors should verify this statement according to the literature.
- (d) Mechanism analysis (Figs. 4, 5): The discussion on the impact of Pb content from "local" to "global" could be more in-depth. It is suggested to briefly discuss this in conjunction with the trends in A-O bond length changes (Figs. 5n-p) and the hybridization of electron clouds between A-O from DFT calculations.
- (e) The P-E loops testing field (110 kV/cm) of the optimum sample is much higher than the charge-discharge test field (46 kV/cm). Why?

3. Expression and Format:

(a) Full terms should be provided upon first mention; for example, "SPE" (superparaelectric) is recommended to be given in full when first appearing in the abstract and introduction.

(b) Ensure accurate citations of all figures and tables in the text. For instance, page 3 mentions "(as shown in Figures a1-a3)," but the actual figure labels in the text are Fig. 1a1-1a3, etc. A thorough check and correction for consistency are needed.

(c) The reference format requires a full-text check. Inconsistencies exist in the formatting of journal name abbreviations, volume, issue, and page numbers in some references (e.g., year 2026 in ref 8, page number formats like e22382/e17944 in refs 23/24, etc.). Please standardize according to the target journal's requirements.

4. Conclusion:

The conclusion could be further condensed to highlight the most core finding of this work: By modulating the local distribution of the highly polar element Pb, "on-demand" enhancement of the polarization response capability in high-entropy materials is achieved, thereby optimizing both energy storage and piezoelectric performance within the same material system. This provides a new paradigm for the functional design of high-entropy ferroelectrics.

Reviewer #2

(Remarks to the Author)

The authors have presented a study characterizing the electrical properties for superparaelectric material systems (BNKLSTZ) in the form of bulk ceramics and their corresponding composite phases with (anti)ferroelectric or paraelectric. Here are some recommendations which can help improve the readability and reproducibility of the work being discussed in the manuscript:

(a) Page 8 (Experimental Validation section, paragraph 2): The authors mention "For an ideal ferroelectric, γ is 1, while for a relaxor ferroelectric, it is 2. As shown in Fig. S12, 20CT, 20BT, and 20PT ceramics all exhibit high γ values of ~ 1.8 , whereas 20PZ has a lower γ of ~ 1.5 . A γ value close to 2 indicates that the introduction of the strongly polar ferroelectric phases BT and PT does not disrupt the relaxor characteristics of the superparaelectric."

Such statements can seem confusing to the reader. Are the authors suggesting the superparaelectric phases being considered in the current study are the same as relaxors? If so, why are the superparaelectric materials not being referred to as relaxors? Relaxors (like PMNPT) are a well-studied group of material systems which have been studied widely for their energy storage properties. The authors should clarify more.

(b) The authors should consider extracting the polarization response as a function electric field and the electrical breakdown strength in a statistically meaningful manner (eg. Weibull distribution function) for their studied sample compositions for a reproduceable and reliable estimate of the capacitive energy storage-related parameters.

(c) The authors should consider presenting the energy density (W_{rec}) as a function of increasing temperature and cyclic fatigue to test the reliability of the material systems as candidates for capacitive energy storage in the main text. Most studies in the literature on capacitive energy storage use such temperature- and fatigue-dependent change in energy storage behavior as an important metric for the figure of merit of the material system for the authors' application of interest.

(d) The authors report a energy storage density of ~ 21 J/cm³ and an efficiency of $\sim 87\%$. There is literature available for material systems with higher energy storage density as well as higher efficiency than that reported by the authors. The authors should consider adding more insight (either through experimental studies or theoretical simulations) on how to improve on the energy storage properties (apart from what has been achieved) for their proposed material system which can guide the readers on the materials design strategies for energy storage related applications for bulk ceramics.

Reviewer #3

(Remarks to the Author)

Excellent work. I recommend "accept with minor corrections".

1. In the Methods section, there is a typo of RAMAN (it is written RAMMA). Authors need to correct it.

2. I could not get the justification for calculating the PDOS of the material in this context. Authors need to justify why it was calculated and how it is contributing to the study.

Version 1:

Reviewer comments:

Reviewer #1

(Remarks to the Author)

The authors have adequately addressed my concerns. The revisions are satisfactory. I recommend this manuscript for acceptance.

Reviewer #2

(Remarks to the Author)

accept the manuscript in its current revised form

Reviewer #3

(Remarks to the Author)

I am satisfied with the authors' responses to the comments made in the revised manuscript.

REVIEWER COMMENTS AND RESPONSES

We sincerely thank the reviewers for their valuable comments and give us another chance to revise our manuscript. The feedback has been extremely helpful. We have made the modifications according to the comments. The revisions and response to reviewers' comments are summarized as follows.

Reviewer #1

This paper proposes an innovative design strategy of "local ferroelectric–global superparaelectric." By introducing the strong ferroelectric phase PbTiO_3 into a high-entropy superparaelectric matrix (BNKLSTZ) and leveraging the coupling between them, an extremely high polarization response under an electric field is achieved, resulting in an ultrahigh recoverable energy storage density ($\sim 21 \text{ J/cm}^3$) and high efficiency ($\sim 87\%$). The research comprehensively employs phase-field simulations, experimental fabrication, multi-scale structural characterization (in-situ XRD, in-situ Raman, atomic-scale STEM, etc.), and first-principles calculations to systematically demonstrate the critical role of the local Pb^{2+} lone pair effect in enhancing local ferroelectric polarization while maintaining the overall superparaelectric structure. The work is systematic and complete, with substantial data, offering significant theoretical and practical guidance for the design of high-performance energy storage ceramics and high-entropy ferroelectric materials. Overall, this is a high-quality study, and I recommend modifications and improvements based on the following specific comments before acceptance.

Response: Thank you very much for reviewing our manuscript and for your highly positive assessment. We have carefully read through your insightful and constructive comments. These comments are highly pertinent and provide crucial guidance for enhancing the quality of our manuscript. Accordingly, we have meticulously revised and optimized the manuscript point by point based on your recommendations. We hope these additions meet your expectations and further enhance the contribution and impact of the manuscript.

Comment 1. Abstract and Introduction:

Response: Thanks for your insightful comments. We greatly appreciate your constructive comments, which have helped us improve the clarity and completeness of our manuscript. Following your advice, we have revised the Abstract and Introduction accordingly.

(1) The abstract could be further refined to explicitly highlight the core advantage of the "local ferroelectric–global superparaelectric" strategy over a pure high-entropy strategy (i.e., addressing the bottleneck of limited polarization (P_m) in high-entropy systems).

Response: Thanks for your insightful comments. We have revised the abstract to clearly state that the "local

ferroelectric–global superparaelectric” design overcomes the limitation of low saturation polarization in conventional high-entropy superparaelectrics by introducing local strong polarization without compromising the global superparaelectric characteristics. In the sections highlighted in yellow on pages 1 of the revised manuscript, the following modifications have been made.

“...In this work, guided by phase field simulations and experimental evidence, we propose a “local ferroelectric–global superparaelectric” design strategy, which introduces local ferroelectric polarization that significantly enhance P_m while preserving the superparaelectric matrix, thereby achieving superior energy storage performance...”

(2) When reviewing how the high-entropy strategy enhances energy storage performance in the Introduction, a more specific explanation of "how the high-entropy effect leads to the formation of polar nanoregions (PNRs)" would clarify the logical chain.

Response: Thanks for your insightful suggestions. We have expanded the Introduction to elaborate on the mechanism by which the high-entropy effect promotes the formation of polar nanoregions (PNRs). Specifically, we explained that the coexistence of multiple cations at equivalent crystallographic sites introduces local compositional disorder, random fields, and lattice distortions, which disrupt long-range ferroelectric order and lead to the formation of nanoscale polar regions. In the sections highlighted in yellow on page 2 of the revised manuscript, the following modifications have been made.

“...The underlying mechanism primarily lies in the coexistence of multiple dissimilar atoms in the same lattice position within the high-entropy system. These cations possess distinct ionic radii, valences, and electronic configurations, which generate significant local compositional disorder, random fields, and lattice distortions, disrupting the long-range ferroelectric order and promote the formation of polar nanoregions (PNRs)...”

(3) The article lacks citations to relevant studies that could support the statements provided related to Fig. 1; for example, " Density functional theory analysis reveals that the presence of the lone pair electrons on Pb is responsible for generating the localized strong polarity, providing the atomic-scale mechanism for the performance enhancement." etc. The authors are asked to cite the original related work that supports your statements.

Response: Thanks for your insightful comments. We have added appropriate reference (Cohen, Nature. 1992) to support the DFT-based discussion on the role of Pb’s lone pair electrons in enhancing the localized strong polarity.

Comment 2. Results and Discussion:

Response: Thank you for your careful review and valuable comments. We have carefully addressed each comment as follows:

(1) Phase-field simulations (Fig. 2): The label "P-E loop" in Fig. 2d4 should be clearly stated as "schematic of the simulated P-E loop characteristics."

Response: Thank you for your careful reviewing. We have revised the figure label accordingly.

(2) Experimental verification (Fig. 3): The test electric field (2 kV/mm) for the P-E loops of different compositions in Fig. 3i-l is relatively low, serving to verify trends. It is recommended to add an explanation in the main text that differences observed under this field are sufficient to support the simulation conclusions, while the subsequently optimized performance is obtained under high breakdown field strengths.

Response: Thank you for this thoughtful observation. We agree that clarifying the purpose of the low-field measurements in Fig. 3i-l will help readers better understand our experimental logic. We have added an explanation in the main text to clarify this point, as shown below.

“...It is worth noting that the P-E loops shown in Figs. 3i-l were measured at a relatively low electric field of 2 kV/mm. This field strength was deliberately chosen to qualitatively verify the trends predicted by phase-field simulations. The results obtained under this low field consistently support the simulation conclusions. Building on this mechanistic understanding, subsequent energy storage performance optimization by designing PT content was conducted under breakdown field...”

In the revised manuscript, these modifications have been highlighted on page 9.

(3) Verification of SPE state: The authors selected BNKLSTZ high-entropy ceramics as the matrix. As shown in Fig. S9a, temperature-related dielectric permittivity indicate that the high-entropy matrix has a T_m far below room temperature (~ -101 °C), confirming its room-temperature superparaelectric state. The authors should verify this statement according to the literature.

Response: Thank you for your rigorous and careful review. We appreciate this important comment regarding the verification of the superparaelectric state. According to the literature, the superparaelectric state in relaxor ferroelectrics is defined as the temperature range between T_m (the temperature of maximum dielectric permittivity) and T_B (the Burns temperature), where polar nanoregions (PNRs) begin to form but remain dynamically fluctuating with minimized coupling and switching barriers¹⁻³. In our original manuscript, we only referred to T_m being far below room temperature (~ -101 °C) to justify the superparaelectric state, which was incomplete. Following your suggestion, we have now analyzed both T_m

and T_B for the BNKLSTZ high-entropy matrix. As shown in Figure R1 (added as Fig. S9b and S16 in the revised Supplementary Information), the Burns temperature T_B was determined from the temperature-related dielectric permittivity fitting, yielding $T_B \sim 100$ °C-150 °C. Therefore, room temperature (20–25 °C) falls well within the range of $T_m < T < T_B$, confirming that the BNKLSTZ-xPT ($x \leq 30$ mol%) ceramics indeed in a superparaelectric state at room temperature. We have added an explanation in the main text to clarify this point, as shown below.

“...We selected BNKLSTZ high-entropy ceramics as the matrix. The superparaelectric state in relaxor ferroelectrics is defined as the temperature range between T_m (the temperature of maximum dielectric permittivity) and T_B (the Burns temperature), where polar nanoregions (PNRs) begin to form but remain dynamically fluctuating with minimized coupling and switching barriers. As shown in Figs. S9a-b, temperature-related dielectric permittivity indicate that the high-entropy matrix has a T_m (~ -101 °C) far below room temperature and T_B (~ 100 °C) above room temperature, confirming its room-temperature superparaelectric state...”

“...The temperature-related dielectric permittivity and loss results (Figs. 4b-d and Figs. S16a-d) are consistent with XRD. When the PT content is ≤ 30 mol%, the T_m peak is below room temperature and T_B is above room temperature, indicating that the ceramics are in a superparaelectric state (Figs. S16e-h)...”

In the revised manuscript, these modifications have been highlighted on pages 8 and 11, and refs [1]-[3] were cited.

Figure R1. $1/\epsilon_r$ -T of BNKLSTZ-xPT ceramics. (a)x=0; (b)x=0.1; (c) x=0.2; (d) x=0.25; (e)x=0.3.

References:

1. Duan, J., et al., High-entropy superparaelectrics with locally diverse ferroic distortion for high-capacitive energy storage. *Nature Communications*, 2024. 15(1): p. 6754.
2. Pan, H., et al., Ultrahigh energy storage in superparaelectric relaxor ferroelectrics. *Science*, 2021. 374(6563): p. 100-104.
3. Chen, L., et al., Local Diverse Polarization Optimized Comprehensive Energy-Storage Performance in Lead-Free Superparaelectrics. *Advanced Materials*, 2022. 34(44): p. 2205787.

(4) Mechanism analysis (Figs. 5, 6): The discussion on the impact of Pb content from "local" to "global" could be more in-depth. It is suggested to briefly discuss this in conjunction with the trends in A-O bond length changes (Figs. 6n-p) and the hybridization of electron clouds between A-O from DFT calculations.

Response: Thank you for this insightful suggestion. We agree that a more in-depth discussion will strengthen our mechanism analysis. Following your advice, we have expanded the discussion in Section (5) to explicitly correlate the trends in A-O bond length changes (Figs. 6n-p) and A-O electron cloud hybridization from DFT calculations with the evolution of polarization behavior. The modification is detailed below.

“...This progressive enhancement in A-O bond length asymmetry and orbital hybridization with increasing Pb content directly correlates with the microstructural evolution observed in STEM (Figs. 6h-j). At low Pb content (25%Pb), the A-O bond length difference and Pb-O hybridization remain spatially confined within isolated polar nanoregions, preserving the global superparaelectric matrix and resulting in enhanced energy storage performance. At high Pb content, the A-O bond length difference and Pb-O hybridization further increases, indicating that strongly distorted Pb-O bonding networks percolate throughout the entire lattice, leading to long-range ferroelectric order and the emergence of macroscopic piezoelectricity...”

In the revised manuscript, these modifications have been highlighted on page 17.

(5) The P-E loops testing field (110 kV/cm) of the optimum sample is much higher than the charge-discharge test field (46 kV/cm). Why?

Response: Thank you for your insightful question regarding the different electric field strengths used in P-E loop and charge-discharge measurements. The P-E hysteresis loops of the optimum 25PT ceramic were measured at the breakdown field strength (110 kV/mm) to evaluate the maximum achievable energy storage density (W_{rec}) of the material. In dielectric energy storage research, P-E loops are typically tested up to the breakdown field to determine the upper limit of energy storage performance, as W_{rec} is calculated by integrating the area between the polarization curve and the polarization axis up to the maximum applied field. This allows us to demonstrate the full potential of our material system. In contrast, the

charge-discharge tests were conducted at a moderate field strength of 46 kV/mm. This choice is consistent with common practice in the literature, where charge-discharge measurements are typically performed at intermediate electric fields (typically in the range of ~10-50 kV/mm for bulk ceramics) to evaluate practical performance metrics relevant to pulsed power applications³⁻⁶. The moderate field of 46 kV/mm used in our charge-discharge tests falls well within the range commonly reported in the literature. This approach allows for a meaningful comparison of our material's pulsed performance with state-of-the-art results while maintaining measurement stability.

References

3. Chen, L., et al., Local Diverse Polarization Optimized Comprehensive Energy-Storage Performance in Lead-Free Superparaelectrics. *Advanced Materials*, 2022. 34(44): p. 2205787.
4. Chen, L., et al., Large Energy Capacitive High-Entropy Lead-Free Ferroelectrics. *Nano-Micro Letters*, 2023. 15(1): p. 65.
5. Liu, H., et al., Local Chemical Clustering Enabled Ultrahigh Capacitive Energy Storage in Pb-Free Relaxors. *Journal of the American Chemical Society*, 2023. 145(35): p. 19396-19404.
6. Xiong, X., et al., Ultrahigh Energy-Storage in Dual-Phase Relaxor Ferroelectric Ceramics. *Advanced Materials*, 2024. 36(48): p. 2410088.

Comment 3. Expression and Format:

Response: Thank you for your careful review and valuable comments. We have carefully addressed each comment as follows:

(1) Full terms should be provided upon first mention; for example, "SPE" (superparaelectric) is recommended to be given in full when first appearing in the abstract and introduction.

Response: Thank you for your careful review. We have revised the abstract and introduction to ensure that all abbreviations are defined upon first appearance. Specifically, "superparaelectric (SPE)" is now provided in full when first mentioned in both the abstract and the introduction. In the sections highlighted in yellow on pages 1 and 2 of the revised draft, the following modifications have been made.

“...which introduces local ferroelectric polarization that significantly enhance P_m while preserving the superparaelectric (SPE) matrix, thereby achieving superior energy storage performance...”

“...often drives the system into a superparaelectric (SPE) state...”

(2) Ensure accurate citations of all figures and tables in the text. For instance, page 3 mentions "(as shown in Figures a1- a3)," but the actual figure labels in the text are Fig. 1a1-1a3, etc. A thorough check and correction for consistency are needed.

Response: We thank the reviewer for pointing out the inconsistency in figure citations. We have thoroughly checked the entire manuscript and corrected all figure references to ensure accuracy and consistency with the actual figure labels (e.g., revising "Figures a1-a3" to "Figs. 1a1-1a3"). All figure and table citations have been verified throughout the text.

(3) The reference format requires a full-text check. Inconsistencies exist in the formatting of journal name abbreviations, volume, issue, and page numbers in some references (e.g., year 2026 in ref 8, page number formats like e22382/e17944 in refs 23/24, etc.). Please standardize according to the target journal's requirements.

Response: Thank you for your careful review. We have carefully reviewed and standardized all references according to the target journal's formatting requirements. Specifically, we have: (i) unified journal name abbreviations, (ii) ensured consistent formatting of volume, issue, and page numbers, and (iii) corrected special cases such as the publication year (e.g., ref. 8) and page number formats (e.g., refs. 23 and 24). A full reference list has been checked and revised accordingly.

Comment 4. Conclusion:

The conclusion could be further condensed to highlight the most core finding of this work: By modulating the local distribution of the highly polar element Pb, "on-demand" enhancement of the polarization response capability in high-entropy materials is achieved, thereby optimizing both energy storage and piezoelectric performance within the same material system. This provides a new paradigm for the functional design of high-entropy ferroelectrics.

Response: Thank you for your valuable comment. We agree that the conclusion should be more concise and highlight the most significant contribution of our work. Following your advice, the revised conclusion is presented below. In the sections highlighted in yellow on pages 18 of the revised manuscript, the following modifications have been made.

“...Moreover, by modulating the distribution of the ferroelectric polarization within a high-entropy superparaelectric matrix, we achieve on-demand enhancement of polarization response, enabling optimization of energy storage and piezoelectric performance in the same material system....”

Reviewer #2

The authors have presented a study characterizing the electrical properties for superparaelectric material systems (BNKLSTZ) in the form of bulk ceramics and their corresponding composite phases with (anti)ferroelectric or paraelectric. Here are some recommendations which can help improve the readability and reproducibility of the work being discussed in the manuscript:

Response: Thank you very much for reviewing our manuscript and for your highly positive assessment. We have carefully read through your insightful and constructive comments. These comments are highly pertinent and provide crucial guidance for enhancing the quality of our manuscript. Accordingly, we have meticulously revised and optimized the manuscript point by point based on your recommendations. We hope these additions meet your expectations and further enhance the contribution and impact of the manuscript.

(a) Page 8 (Experimental Validation section, paragraph 2): The authors mention "For an ideal ferroelectric, γ is 1, while for a relaxor ferroelectric, it is 2. As shown in Fig. S12, 20CT, 20BT, and 20PT ceramics all exhibit high γ values of ~ 1.8 , whereas 20PZ has a lower γ of ~ 1.5 . A γ value close to 2 indicates that the introduction of the strongly polar ferroelectric phases BT and PT does not disrupt the relaxor characteristics of the superparaelectric." Such statements can seem confusing to the reader. Are the authors suggesting the superparaelectric phases being considered in the current study are the same as relaxors? If so, why are the superparaelectric materials not being referred to as relaxors? Relaxors (like PMNPT) are a well-studied group of material systems which have been studied widely for their energy storage properties. The authors should clarify more.

Response: Thank you for this insightful comment. We appreciate the opportunity to clarify the relationship between superparaelectric states and relaxor ferroelectrics in our manuscript.

According to the established literature, the superparaelectric state is a specific temperature regime within relaxor ferroelectrics, defined as the temperature range between T_m (the temperature of maximum dielectric permittivity) and T_B (the Burns temperature)¹⁻³. In this regime, polar nanoregions (PNRs) exist but remain dynamically fluctuating with minimized coupling and switching barriers, enabling rapid polarization response with significantly suppressed hysteresis. In our manuscript, employing the terminology of the superparaelectric state enables a more accurate description of the material system discussed in the manuscript. Following your comment, we have revised the relevant paragraph to clearly articulate this relationship. The modified text is provided below.

“...We selected BNKLSTZ high-entropy ceramics as the matrix. The superparaelectric state in relaxor ferroelectrics is defined as the temperature range between T_m (the temperature of maximum dielectric permittivity) and T_B (the Burns temperature), where polar nanoregions (PNRs) begin to form but remain

dynamically fluctuating with minimized coupling and switching barriers...”

In the revised manuscript, the corresponding text has been revised on page 8.

References:

- 1 Duan, J. et al. High-entropy superparaelectric with locally diverse ferroic distortion for high-capacitive energy storage. *Nature Communications* 15, 6754 (2024). <https://doi.org/10.1038/s41467-024-51058-6>
- 2 Pan, H. et al. Ultrahigh energy storage in superparaelectric relaxor ferroelectrics. *Science* 374, 100-104 (2021). <https://doi.org/doi:10.1126/science.abi7687>
- 3 Chen, L. et al. Local Diverse Polarization Optimized Comprehensive Energy-Storage Performance in Lead-Free Superparaelectrics. *Advanced Materials* 34, 2205787 (2022). <https://doi.org/https://doi.org/10.1002/adma.202205787>

(b) The authors should consider extracting the polarization response as a function electric field and the electrical breakdown strength in a statistically meaningful manner (eg. Weibull distribution function) for their studied sample compositions for a reproduceable and reliable estimate of the capacitive energy storage-related parameters.

Response: Thank you for this valuable comment. We agree that employing Weibull distribution analysis is essential for obtaining reproducible and reliable estimates of capacitive energy storage parameters, particularly the breakdown strength (E_b), which directly influences the achievable energy storage density.

(1) In the revised manuscript, we have performed Weibull statistical analysis on the breakdown strength for the 25PT ceramic. The characteristic breakdown strength obtained from Weibull fitting is ~ 111.5 kV/mm, as shown in Fig. R2a, which closely matches the experimentally applied field of 110 kV/mm. The high shape parameter (β) ~ 19.4 further confirms the reliability of the breakdown data. These results demonstrate that the reported energy storage density of ~ 21 J/cm³ is statistically reproducible and reliable.

(2) We also evaluated the sample-to-sample variation in polarization response as a function of electric field. As shown in Fig. R2b, additional four 25PT ceramics exhibit nearly identical P_m and P_r values as a function of electric field. Under the characteristic breakdown field of ~ 110 kV/mm, these samples yield consistent energy storage densities of 21 ± 1 J/cm³ and efficiencies of $87 \pm 1\%$, further confirming the excellent reproducibility of the energy storage performance.

Together, these statistical analyses validate the reliability of our reported capacitive energy storage parameters. In the revised manuscript, on page 12, we have added the following description, and Figure R2 has been included as Supplementary Material Figure S20.

“...To verify the statistical reliability of the reported energy storage performance, we performed Weibull analysis on the breakdown strength of the 25PT ceramic. As shown in Fig. S20a, the breakdown data exhibit

excellent linear fitting to the Weibull distribution, yielding a characteristic breakdown strength of $E_b \sim 111.5$ kV/mm with a high shape parameter of $\beta \sim 19.4$. We further evaluated the sample-to-sample reproducibility of the polarization response. As shown in Fig. S20b, four independently prepared 25PT samples exhibit nearly identical P_m and P_r values as a function of electric field. Under the characteristic breakdown field of 110 kV/mm, these samples yield consistent energy storage densities of 21 ± 1 J/cm³ and efficiencies of $87 \pm 1\%$. These results demonstrate that the ultrahigh energy storage performance of the 25PT ceramic is both reproducible and reliable...”

Figure R2 (a) Weibull analysis of the 25PT ceramics; (b) Polarization response as a function of electric field for different 25PT samples.

(c) The authors should consider presenting the energy density (W_{rec}) as a function of increasing temperature and cyclic fatigue to test the reliability of the material systems as candidates for capacitive energy storage in the main text. Most studies in the literature on capacitive energy storage use such temperature- and fatigue-dependent change in energy storage behavior as an important metric for the figure of merit of the material system for the authors' application of interest.

Response: Thank you for this important suggestion regarding the presentation of temperature-dependent and fatigue-dependent energy storage performance. We fully recognize that for practical applications, the stability of energy storage performance under thermal and electrical cycling is equally critical. Following your recommendation, as shown in Fig. R3i-j, we have presented the energy density (W_{rec}) as a function of increasing temperature and cyclic fatigue to the main text (Fig. 4) and added a discussion on the temperature and fatigue reliability of our optimum 25PT ceramic. The revised manuscript now includes:

“...For practical applications, the stability of energy storage performance under varying thermal and electrical conditions is equally important. As shown in Figs. 4i1-i2, the 25PT ceramic exhibits excellent

temperature stability in the range of 20–160 °C at 70 kV/mm, with $W_{rec} \sim 11.3 \pm 0.6 \text{ J/cm}^3$ and $\eta \sim 87.2 \pm 1.7\%$, demonstrating minimal degradation even at elevated temperatures. The fatigue test (Figs. 4j1-j2) reveals exceptional cycling reliability, with W_{rec} and η exhibiting negligible variation after 10^7 charge-discharge cycles ($W_{rec} \sim 11.8 \pm 0.1 \text{ J/cm}^3$, $\eta \sim 87.9 \pm 0.8\%$)...

“Figure R3 ... (i) P-E hysteresis loops of the BNKLSTZ-25PT ceramic at 70 kV/mm over a wide range of (i1) the temperature from 20 to 160 °C with (i2) the corresponding W_{rec} and η ; (j) P-E hysteresis loops of the BNKLSTZ-25PT ceramic at 70 kV/mm over a wide range of (j1) cycle number from 1 to 10^7 with (j2) the corresponding W_{rec} and η ..”

In the revised manuscript, the above text has been updated on page 12, and Figure 4 has been revised to Figure R3.

(d) The authors report an energy storage density of $\sim 21 \text{ J/cm}^3$ and an efficiency of $\sim 87\%$. There is literature available for material systems with higher energy storage density as well as higher efficiency than that reported by the authors. The authors should consider adding more insight (either through experimental studies or theoretical simulations) on how to improve on the energy storage properties (apart from what has been achieved) for their proposed material system which can guide the readers on the materials design strategies for energy storage related applications for bulk ceramics.

Response: We sincerely thank the reviewer for the insightful comment, which has been highly enlightening for our future research. In previous manuscripts, we mainly focus on investigating the influence of

ferroelectric polarization on the structure and properties of the high-entropy superparaelectric. As the reviewer suggested, we have added further insights into how the energy storage properties can be potentially enhanced, based on the mechanisms revealed in our study. Specifically, according to the $W_{\text{rec}} = \int_{P_r}^{P_m} E dP$, the key to further improving performance lies in the parameters: P_m-P_r , and E .

(1) Enhancing the (P_m-P_r): as shown in Figure R4, we have supplemented **additional theoretical calculations** to analyze how to further enhance P_m-P_r . Our further phase-field simulations indicate that the critical factor is not merely the introduction of a ferroelectric polarization, but rather the control of its distribution. As illustrated in Figures R4a and R4c, at a given concentration, a more concentrated distribution of the ferroelectric polarization renders it less susceptible to the influence of the superparaelectric matrix. Consequently, it becomes difficult to maintain a near-zero polarization (large P_r) after the removal of the electric field, thereby deteriorating the energy storage density and efficiency. Conversely, as illustrated in Figures R4b~c, a more dispersed distribution of the ferroelectric polarization enhances the coupling between the ferroelectric and superparaelectric regions, facilitating the return of polarization to its initial state. This leads to a reduced P_r and improved energy storage density and efficiency. Based on the current pressureless sintering process, for the BNKLSTZ-xPT system, the optimal composition lies near $x = 0.25$. Future work could focus on fine-tuning the composition within the range of $x = 0.25$ to 0.4, or employing novel sintering techniques to control the size and distribution of ferroelectric polarization. This approach holds the potential to further enhance the (P_m-P_r) value, thereby improving the energy storage density and efficiency.

(2) Enhancing the E_b : as presented in the response to Comment (b), the current ultrahigh W_{rec} of 21 J/cm³ is achieved at 110 kV/mm, which is the statistic breakdown field strength. As illustrated in figures R5a1-a2, at higher field strength of 120 kV/mm, the material exhibits an increased energy storage density of ~23 J/cm³ while maintain efficiency of ~85%. These results indicate that further engineering of the microstructure, such as suppressing elemental volatilization (i.e., Pb), could inhibit electrical trees and enhance E_b , thereby potentially increasing energy storage performances.

(3) Experimental study: Here, as shown in Figure R5b1, we employed the double-crucible method to suppress the volatilization of Pb, thereby further enhancing the performance of the 25PT ceramics. As illustrated in Figure R5b2, at an electric field of 122 kV/mm, both the energy storage density and efficiency of the 25PT ceramics were improved, reaching 24.8 J/cm³ and 88%, respectively. The suppression of Pb volatilization is likely to have promoted a more uniform distribution of ferroelectric polarization and a denser microstructure, thereby enhancing P_m-P_r and E_b , which ultimately contributed to the improved energy storage performance.

The above discussion is provided in Supplementary Material Note 5, and Figures R4-R5 has been incorporated into the Supplementary Materials as Fig. S34 and Fig. S35. Moreover, on pages 17-18 of the revised manuscript, we have included further insights as follows,

“...Based on the mechanistic insights obtained in this work, further enhancement of energy storage performance can be anticipated by optimizing the distribution of the ferroelectric polarization to maximize the polarization difference ($P_m - P_r$), and by engineering the microstructure to increase the breakdown strength E_b ((Supplementary Note 5 and Figs.S34~S35). These strategies, guided by the "local ferroelectric–global superparaelectric" design principle, offer a promising pathway toward even higher energy storage performances in high-entropy ferroelectric ceramics...”

Figure R4 Phase-field simulation on the effects of different ferroelectric polarization distributions on energy storage performance at the same content: (a) Concentrated distribution of ferroelectric polarization; (b) Relatively uniform distribution of ferroelectric polarization; (c) Simulated P-E loops; (d) P-E loops of 25 PT ceramics under 120 kV/mm.

Figure R5 (a) P-E loops under 120 kV/mm of 25 PT ceramics prepared by single-crucible method; (b) P-E loops under 122 kV/mm of 25 PT ceramics prepared by double-crucible method.

Reviewer #3

Excellent work. I recommend "accept with minor corrections".

Response: Thank you very much for reviewing our manuscript and for your highly positive assessment. We have carefully read through your insightful and constructive comments. These comments are highly pertinent and provide crucial guidance for enhancing the quality of our manuscript. Accordingly, we have meticulously revised and optimized the manuscript point by point based on your recommendations. We hope these additions meet your expectations and further enhance the contribution and impact of the manuscript.

1. In the Methods section, there is a typo of RAMAN (it is written RAMMA). Authors need to correct it.

Response: We thank the reviewer for pointing out the typo. We have carefully checked the manuscript and corrected "RAMMA" to "Raman" in the Methods section. The revised sentence now reads:

"...The in situ Raman spectra were obtained by Raman spectromete ..."

In the revised manuscript, the above text has been updated on page 20.

2. I could not get the justification for calculating the PDOS of the material in this context. Authors need to justify why it was calculated and how it is contributing to the study.

Response: We thank the reviewer for this insightful comment. The partial density of states (PDOS) calculations was performed to elucidate how the electronic configurations of different elements influence structure distortion at the atomic scale, which is central to understanding the performance enhancement observed in our study. Specifically, the PDOS provides critical evidence for the role of Pb's lone pair electrons. In Pb-containing structure, we observe significant hybridization between Pb 6s and O 2p orbitals, indicating the active lone pair effect. This hybridization creates an asymmetric electron distribution around Pb, which promotes its off-center displacement within the perovskite cage and enhances structure distortion. In contrast, A-site elements such as Ba, which lack lone pair electrons, show no such hybridization and thus contribute less to local polarization enhancement. This explains the mechanism of the significant changes in structure and properties caused by the introduction of Pb from the perspective of electronic configuration. We have made modifications to the revised manuscript in Page 8, as detailed below.

"...The structural features and underlying electronic origins of these components were analyzed using density functional theory calculations (Supplementary Note 3 and Fig. S6). Specifically, structural relaxation was performed to evaluate the type and degree of structure distortion in each compound, while charge density and partial density of states (PDOS) analyses were employed to uncover the electronic configuration origins of these structural variations..."

"...As shown in Fig. 3b, BaTiO₃ has a tolerance factor of ~1.06, which favors the formation of a

ferroelectric tetragonal structure. This distortion is achieved through hybridization between Ti and O (Figs. S7a-b and S8a-b), where Ti possesses unoccupied d orbitals that facilitate orbital mixing with O 2p states, enabling its off-center displacement and resulting in a relative cation-anion shift along the [001] direction. As shown in Fig. 3c, although PbTiO_3 has a relatively lower tolerance factor of ~ 1.02 , the presence of the lone pair on the Pb ion (Figs. S7c-d and S8c-d) significantly enhances its ferroelectric distortion, far exceeding that of BaTiO_3 ...

Once again, many thanks for reviewers' valuable feedback and constructive comments. We appreciate the time and effort you have dedicated to reviewing our manuscript. Your insights have been instrumental in improving the quality of our work. We hope our revisions meet your expectations and further enhance the contribution and impact of the manuscript.

Best regards,

Dou Zhang

State Key Laboratory of Powder Metallurgy

Central South University

Changsha, Hunan 410083

China

REVIEWER COMMENTS AND RESPONSES

We sincerely thank the reviewer for their valuable comment. The response to reviewers' comment is summarized as follows.

Reviewer #1 (Remarks to the Author):

The authors have adequately addressed my concerns. The revisions are satisfactory. I recommend this manuscript for acceptance.

Response: We are pleased to hear that the revisions have addressed your comments, and we appreciate the time and effort that the reviewer has invested in evaluating our manuscript.

Reviewer #2 (Remarks to the Author):

Accept the manuscript in its current revised form.

Response: We are pleased to hear that the revisions have addressed your comments, and we appreciate the time and effort that the reviewer has invested in evaluating our manuscript.

Reviewer #3 (Remarks to the Author):

I am satisfied with the authors' responses to the comments made in the revised manuscript.

Response: We are pleased to hear that the revisions have addressed your comments, and we appreciate the time and effort that the reviewer has invested in evaluating our manuscript.

Best regards,

Prof. Dou Zhang

Email: dzhang@csu.edu.cn

State Key Laboratory of Powder Metallurgy, Central South University, Changsha, Hunan 410083, China.